# Hygroscopic growth of water soluble organic carbon isolated from atmospheric aerosol collected at U.S. national parks and Storm Peak Laboratory

Nathan F. Taylor[1], Don R. Collins[1], Douglas H. Lowenthal[2], Ian B. McCubbin[2], A. Gannet Hallar[2,5], Vera Samburova[2], Barbara Zielinska[2], Naresh Kumar[3], Lynn R. Mazzoleni[4]

[1]Department of Atmospheric Sciences, Texas A&M University, College Station, Texas, U.S.A.
[2]Division of Atmospheric Sciences, Desert Research Institute, Reno, Nevada, U.S.A.
[3]Electric Power Research Institute, Palo Alto, California, U.S.A.
[4]Atmospheric Science Program, Michigan Technological University, Houghton, Michigan, U.S.A.
[5]Department of Atmospheric Sciences, University of Utah, Salt Lake City, Utah, U.S.A.

*Correspondence to*: Don Collins (dcollins@tamu.edu)

**Abstract.** Due to the atmospheric abundance and chemical complexity of Water Soluble Organic Carbon (WSOC), its contribution to the hydration behavior of atmospheric aerosol is both significant and difficult to assess. For the present study, the hygroscopicity and CCN activity of isolated atmospheric WSOC particulate matter was measured without the compounding effects of common, soluble inorganic aerosol constituents. WSOC was extracted with high purity water from daily high-volume PM2.5 filter samples and separated from water soluble inorganic constituents using solid phase extraction. The WSOC filter extracts were concentrated and combined to provide sufficient mass for continuous generation of the WSOC-only aerosol over the combined measurement time of the tandem differential mobility analyzer and coupled scanning mobility particle sizer / CCN counter used for the analysis. Aerosol samples were taken at Great Smoky Mountains National Park during the summer of 2006 and fall-winter of 2007–08; Mount Rainier National Park during the summer of 2009; Storm Peak Laboratory (SPL) near Steamboat Springs, Colorado, during the summer of 2010; and Acadia National Park during the summer of 2011. Across all sampling locations and seasons, the hygroscopic growth of WSOC samples at 90% RH, expressed in terms of the hygroscopicity parameter, $\kappa$, ranged from 0.05 – 0.15. Comparisons between the hygroscopicity of WSOC and that of samples containing all soluble materials extracted from the filters implied a significant modification of the hydration behavior of inorganic components, including decreased hysteresis separating efflorescence and deliquescence and enhanced water uptake between 30 and 70% RH.

*Keywords:* aerosol, organic aerosol, ambient aerosol, hygroscopic growth, hygroscopicity, cloud condensation nuclei, visibility, aerosol hydration, TDMA, WSOC, CCN

# 1 Introduction

Interactions with atmospheric water are central to the impacts of aerosols on human health (Vu et al., 2015), climate (IPCC, 2014) and visibility (Malm and Pitchford, 1997). The tendency of an aerosol to take up water in sub-saturated conditions—its hygroscopicity—directly affects its impact on climate and visibility by modifying the efficiency with which it scatters and absorbs radiation (Tang, 1996). The presence of an aqueous phase in an aerosol affects its aging in the atmosphere, generally increasing the uptake of reactive and soluble trace gases and enhancing aerosol growth (Herrmann et al., 2015; Carlton and Turpin, 2013), significantly impacting all of its atmospheric roles. Likewise, atmospheric aerosols play a critical role in earth's climate as cloud condensation nuclei (CCN) (IPCC, 2014), while cloud processing, in turn, may dramatically modify the size distribution and chemical composition of aerosols (Herrmann et al., 2015).

Water soluble organic carbon (WSOC) is well-known to constitute a major fraction of atmospheric aerosols. Measurements at various locations and seasons, including the measurements associated with this report, have shown WSOC to constitute from 20 to >90% of the total aerosol organic carbon (OC) (Du et al., 2014; Lowenthal et al., 2014; Anderson et al., 2008; Sullivan et al., 2004; Saxena and Hildemann, 1996). Atmospheric WSOC correlates strongly with secondary organic aerosol (SOA), but also has sources in primary biogenic aerosol such as pollen (Miyazaki et al., 2012) and that from biomass

burning (Timonen et al., 2013).  Sun et al. (2011) found that biogenic SOA (characterized by mass spectra) constituted ~75% of WSOC during the summer at rural sites in the eastern U.S., while biomass burning-like aerosol dominated during winter months.  Finally, studies applying dual-isotope, radiocarbon analysis to WSOC indicate that modern carbon predominates in WSOC, but that fossil-carbon in aerosols can also contribute significantly (~20% of WSOC) in polluted air masses (Kirillova

et al., 2014; Kirillova et al., 2013; Miyazaki et al., 2012; Kirillova et al., 2010).

The atmospheric prevalence of WSOC anticipates a significant WSOC contribution to global, water-modulated aerosol impacts; yet assessing its impact is similarly challenging to modeling global SOA (Hallquist et al., 2009).  Moreover, far fewer measurements of WSOC properties have been made to support such assessments and validate models than to support

the efforts to assess global SOA.  This report of ambient WSOC hygroscopicity and CCN activity, from a series of five, month-long studies at remote continental sites in the U.S., helps bridge the gap between WSOC prevalence and the impact of WSOC on ambient aerosol properties.

Most methods of analyzing atmospheric aerosol-phase WSOC begin with the collection of a bulk sample of soluble aerosol

material either using a Particle-into-Liquid Sampler (PILS) or by extracting the soluble material from aerosol collected on filters (Psichoudaki and Pandis, 2013).  Thus, the designation of organic material as water soluble has been called 'operational', defined by the method of dissolving the material (Psichoudaki and Pandis, 2013).  These samples of Water Soluble aerosol Material (WSM) are frequently dominated by common inorganic compounds such as sulfate, nitrate, and ammonium. Extensive analysis of WSOC can be conducted on such mixtures. Total Organic Carbon (TOC) instruments can

retrieve WSOC mass concentration (Chow et al., 2004), and chemical speciation can be determined using a wide range of analytical instruments. The hygroscopicity and CCN activity of WSOC can then be inferred by measuring those properties for WSM and estimating the contribution of WSOC from chemical composition and the well-known behavior of soluble inorganic components (Cerully et al., 2015; Kristensen et al., 2012).  Using this approach, Guo et al. (2015) reported an average contribution by WSOC of 35% to particulate water across several sites in the southeastern U.S. However, the

calculations required to subtract the inorganic contribution introduces uncertainty due to the complexity of WSOC composition, limitations of chemical analysis, non-ideal mixtures, and multiplicity of aerosol phases (Hodas et al., 2015). Alternatively, the physical characteristics of purely organic, laboratory-generated aerosols are frequently measured (Frosch et al., 2013; Engelhart et al., 2011; Frosch et al., 2011; Cruz and Pandis, 1997) to inform estimates of ambient WSOC properties and the mechanisms by which WSOC is formed.   However, modeling is required to relate such data to ambient

WSOC.

The measurements of WSOC hygroscopicity and CCN reported here rely on a third approach—isolating WSOC from the inorganic compounds in a WSM sample.  Various methods of isolating WSOC exist, as reviewed by Duarte and Duarte (2011) and Sullivan and Weber (2006), capable of retaining from 50%−90% of sample WSOC with less than 5% retention of

inorganic compounds. To the authors' knowledge, this approach to quantifying WSOC hydration properties has not been widely employed. Notable exceptions are Gysel et al. (2004); Asa-Awuku et al. (2008); and Suda et al. (2012). The earlier work, by Gysel et al. (2004) carried out a bulk separation of organic matter from WSM extracts of filter-collected ambient aerosol. This isolated organic matter was then re-aerosolized and its hygroscopicity analyzed with a Tandem Differential

Mobility Analyzer (TDMA). They reported hygroscopic growth factors, GF, from 1.08−1.17 at 90% relative humidity, RH, corresponding to a hygroscopicity parameter, $\kappa$ (Petters and Kreidenweis, 2007), from ~0.03−0.08. Asa-Awuku et al. (2008) used similar methods not only to 'desalt' samples of biomass burning aerosol, but also to isolate hydrophilic and hydrophobic WSOC—all for the purpose of determining how each fraction impacted aerosol hygroscopic growth within a multi-parameter, Köhler-theory based framework. The approach of Suda et al. (2012) is quite dissimilar, with extracts from

filter samples of smog chamber-produced aerosol fractionated by reversed-phase high-performance liquid chromatography (HPLC). The HPLC eluate was continuously atomized and analyzed using a Droplet Measurement Technologies CCN counter (CCNc). The approach described here bears many similarities to that of Gysel et al. (2004), but captured a larger fraction of the WSOC. Moreover, the present work is distinguished by its focus on connections between the solubilities of organic and inorganic compounds present in the same particles.

**1.1 Study details**

This paper reports measurements of the hygroscopicity and CCN activity of WSOC isolated from aerosol samples collected during a series of five, month-long field campaigns, as detailed in Table 1. The first four campaigns were part of a larger project focused on sources of visibility degradation in U.S. national parks. The measurements were designed to assess the contribution of WSOC to aerosol water content and the resulting impact on visibility. These studies took place at Great

Smoky Mountains (GRSM) National Park in eastern Tennessee during the summer of 2006 and fall-winter of 2007−08, and at Mount Rainier (MORA) and Acadia (ACAD) national parks during the summers of 2009 and 2011, respectively. The primary results from these studies were reported by Lowenthal et al. (2009, 2015) and Taylor et al. (2011). A similar study took place at Storm Peak Laboratory (SPL) near Steamboat Springs, Colorado, in the summer of 2010. The measurements of WSOC at SPL were part of a closure experiment attempting to estimate WSOC hygroscopicity from highly detailed

chemical speciation and were variously reported by Hallar et al. (2013), Samburova et al. (2013), and Mazzoleni et al. (2012). This paper goes beyond those prior analyses to highlight the hydration behavior of ambient WSOC, especially the complementary enhancement of aerosol water uptake in internal mixtures of WSOC and common inorganic aerosol components.

**2 Methodology**

The analysis of WSOC for the five field studies reported here had three major steps: collection of daily, high volume, PM2.5 filter samples; laboratory-based isolation of WSM and WSOC; and analysis of the hygroscopicity and CCN activity of

aerosol generated from those isolated fractions. We provide a brief synopsis of the steps but direct the reader elsewhere for detailed descriptions.

## 2.1 PM 2.5 filter samples

PM2.5 filter samples were taken during five month-long studies. The first two studies, during the summer of 2006 and winter of 2007-2008, were conducted at the Look Rock Air Quality Station (35.633°N, 83.941°W, 806 m asl). Look Rock is in eastern Tennessee, situated on a ridge (~480 m) overlooking Great Smoky Mountains National Park (GRSM) to the south-east and rural farmland of the Tennessee River valley to the north-west. The third study took place near the southeast border of Mount Rainier National Park (MORA) in the state of Washington (46.758°N, 122.124°W, 426 m asl) during the summer of 2009. The site was in a remote valley, surrounded by heavily-logged coniferous forest. The fourth study, during the summer of 2010, took place at Storm Peak Laboratory (SPL), situated on a high ridge in remote northwestern Colorado (40.455°N, 106.744°W, 3214 m asl). Due to its elevation, SPL provides access to the free troposphere and is influenced by both local and distant sources (Hallar et al., 2013). The final study, during the summer of 2011, was conducted on the Schoodic Peninsula in Maine, which lies west of Acadia National Park (ACAD) (44.341°N, 68.060°W, 20 m asl), across Bar Harbor. The coast of Maine near Acadia National Park is mostly rural, while the interior is sparsely inhabited coniferous forest.

Four high-volume (~1100 L/min) PM2.5 filter samplers were operated during each project, as described by Lowenthal et al. (2009). The filters were Teflon-impregnated glass fiber (TIGF) except during the first 12 days of the summer GRSM study, when Zefluor Teflon membrane filters were used. The filters were pre-cleaned by sonication in methanol and dichloromethane. The sample period was 24 hours. Filters were collected daily from each sampler during all projects and were kept refrigerated during storage and shipping after sampling and prior to analysis. Filters were typically processed within three months of the completion of the respective measurement campaign.

## 2.2 WSOC sample production

High molecular weight WSOC, referred to as Humic Acid-Like Substances (HULIS), was isolated from the filters as described in Lowenthal et al. (2009, 2015), following the method of Duarte and Duarte (2005). First, WSM was extracted from each daily set of 4 high-volume filters by sonication in 250 mL of ultrapure water. The WSM solution was passed through a PTFE membrane filter and then concentrated to 15−20 mL using a rotovap under gentle vacuum. WSOC was then isolated from the WSM using XAD-8 and XAD-4 macro-porous resins/chromatography columns: the WSM was acidified and applied to the XAD-8 and XAD-4 columns in series, the columns were rinsed with ultrapure water to remove inorganic material, and the WSOC was eluted from the columns using a mixture of water and methanol. The WSOC eluate was then concentrated by drying and combined with filter extracts from other days to produce sufficient samples for hygroscopicity and CCN activity analysis. After concentration and combination of consecutive daily samples, 5−8 samples of isolated

WSOC material remained for each month-long study. The filter sampling days associated with each sample are given in the data supplement.

Extraction with two resins, rather than one, increased the molecular weight range of retained WSOC components. XAD-8 material is traditionally used to isolate humic-like, high molecular weight substances while the XAD-4 column has been demonstrated to retain lower molecular weight, hydrophilic organic compounds. The use of XAD-4 likely resulted in more complete and representative WSOC extracts when compared with single column approaches. However, a significant fraction of lower molecular weight organic acids, sugars, and alcohols—likely more hygroscopic than HULIS—were not retained by either XAD column (Samburova et al., 2013).

Chemical analyses of inorganic ions and OC were conducted at various stages of the extraction process. Filter captured sulfate, nitrate, and chloride were measured using ion chromatography (IC) (Dionex DX 3000); ammonium was measured using automated colorimetry (Astoria 301A analyzer) (Lowenthal et al. 2009; Samburova et al., 2013). Similar IC and colorimetry measurements were conducted on the extracts (Lowenthal et. al 2009) and to determine the "sulfate cleaning efficiency" reported in Table 1. Filter and extract OC and organic mass (OM) were determined using thermo-optical reflectance (TOR) (Chow et al., 2004) and/or total organic carbon (TOC) (OI Analytical Aurora 1030W TOC Analyzer) methods, which showed good agreement (Lowenthal et al., 2015). During the SPL study, dissolved WSOC was measured using only TOC (Shimadzu Model TOC-VCSH). During the four national park studies, measurement uncertainties for OC, sulfate, nitrate, and ammonium were 12, 5, 26, and 5% (Lowenthal et al., 2015). The OM to OC ratio was evaluated by depositing and drying isolated WSOC sample material on pre-fired quartz filter punches, and combining TOR carbon analysis with before-and-after measurements of the punch mass; a standard value of 1.8 was employed during all studies except SPL, for which 2.1 was assumed. The performance of the WSOC extraction, summarized as WSOC Recovery in Table 1, was evaluated by before-and-after measurements of dissolved OC (Lowenthal et al. 2009). More extensive chemical analysis was conducted on the extracts from samples collected during the SPL project, including detailed speciation of WSOC and IC measurements of crustal ions (using a Dionex CS16 column) in WSM samples (Samburova et al., 2013).

### 2.3 WSOC hygroscopicity and CCN activity analysis

The hygroscopicity and CCN activity of the prepared samples from each study were evaluated within several months of their extraction. The samples were continuously refrigerated, which effectively preserved their properties. A remnant of a GRSM Summer sample was re-analyzed alongside the GRSM Winter samples over a year after its first analysis and showed little change in hygroscopicity. WSOC hygroscopicity analysis was conducted using a TDMA largely identical to that described by Gasparini et al. (2006), configured as depicted in Fig. 1. CCN activity was measured using the same TDMA, operating as

a Scanning Mobility Particle Sizer (SMPS), paired with a DMT CCN-100 counter (CCNc) in the configuration described by Frank et al. (2006).

## 2.4 Sample aerosol generation

The isolated WSOC samples were aerosolized using a TSI 3076 atomizer configured to recirculate overspray. Several precautions were necessary due to the limited sample material available (~5 mg dissolved in ~15 mL of water). First, because the atomizer consumed ~45 mL of sample solution over the course of the RH-scanning TDMA measurements (~3 h) and the SMPS-CCNc measurements (~1.5 h), it was necessary to dilute the ~15 mL samples with ~40 mL of ultrapure water. These very dilute samples were susceptible to contamination from residues in the atomizer assembly and soluble gases in the compressed air stream. To minimize contamination, the atomizer assembly and reservoir were thoroughly purged with ultrapure water between measurements and with particular care following its use with ammonium sulfate for instrument calibration. Further, the compressed air stream was scrubbed using a HEPA filter and canisters containing silica gel desiccant and activated carbon. Contamination was assessed by contrasting the size distribution produced by the atomizer with and without the extracts. Under consistent conditions, the atomizer will produce a consistent droplet size distribution. The resultant dry particle size distribution reflects the concentration of solute in those droplets; a thousand-fold increase in the dissolved concentration of the atomizer solution translates to a ten-fold shift in the diameter of particles produced. Before each sample was placed into the atomizer, the particle size distribution produced from atomizing ultrapure water was checked to ensure it was roughly 1000 times less concentrated than the sample solution and remained consistent for ~30 minutes. The atomizer was re-cleaned if it failed this test.

## 2.5 TDMA and SMPS-CCNc operation

The hygroscopicity of each sample was evaluated in two sets of TDMA measurements: a series to detect the deliquescence of an initially desiccated (~15% RH) aerosol and a series to detect the efflorescence of an initially hydrated (RH ~90%) aerosol. In both, the first DMA column was operated to select dry particles with electrical mobility diameter of 0.07 µm, roughly corresponding to the peak in number concentration of the dried atomizer-generated aerosol. The second DMA column was operated in conjunction with the condensation particle counter (CPC) as a traditional SMPS, capturing the response of this roughly monodisperse aerosol to the humidification and drying processes occurring between the two DMA columns at ~90s intervals. During the first series—the "pre-desiccated" or "deliquescence" scan—the dry, monodisperse aerosol bypassed the humidifying Perma Pure PD-Series Nafion tube bundle as depicted in Fig. 1. The aerosol was next directed through the RH controlled Nafion tube bundle, which was used to track a gradually descending RH setpoint from ~90% to ~25% RH over the course of ~90 minutes. Thus, this configuration mapped the hygroscopic growth response of an initially desiccated aerosol to capture deliquescence behavior. The second series—the "pre-hydrated" or "efflorescence" scan—mapped the hygroscopic growth of an initially hydrated aerosol to detect distinct efflorescence transitions and meta-

stable hydration states. During the second series, the monodisperse aerosol passed through the humidifying Nafion tube bundle, while the controlled Nafion tracked an RH setpoint that gradually increased from ~25% to ~90% RH.

Each series of TDMA measurements resulted in a series of distributions in which the location (representing hydrated particle
size) of a single, narrow mode reflected the hygroscopic growth of the sample. Each distribution was condensed to a single parameter, growth factor, or GF(RH), which is the ratio of the particle size detected by the second DMA/CPC to the dry particle diameter selected in the first DMA column (here 0.07 µm). The basic GF(RH) curves for each sample from each study are included in the data supplement. Growth factor is an intuitive metric of hygroscopicity, but we primarily discuss these results using the hygroscopicity parameterization, $\kappa(RH)$, of Petters and Kreidenweis (2007). GF(RH) translates
directly into $\kappa(RH)$.

(1)

$$\kappa(RH) = ([GF(RH)]^3 - 1)\frac{\left(\exp\left(\frac{A}{GF(RH) \cdot D_p}\right) - RH\right)}{RH}$$

(2)

$$A = \frac{4 \cdot \sigma_{s/a} \cdot MW_{water}}{R \cdot T \cdot \rho_{water}}$$

Here, $\sigma_{s/a}$ is assumed equal to 0.072 J·m$^{-2}$, the surface tension of pure water. $D_p$, the dry diameter, is chosen to reflect the minimum measured over the range in RH, rather than the size selected by the first DMA column (0.07 µm), to correct for a measurement artifact described in Sect. 3.6 below. These particles are assumed to be spherical. Among the advantages of $\kappa(RH)$ is that it is more closely a bulk property of the material; for an ideal solution, $\kappa(RH)$ is independent of RH and proportional to the molecular volume of the solute.

The CPC count array was inverted with respect to the second DMA only; the resulting distributions retained the natural breadth and multiply charged particles contribution from the aerosol population selected by the first DMA. Neither significantly impacts the results. The relative concentration of multiply charged particles is expected to be relatively low, as the generated aerosol distribution was quite narrow and the selected size near its peak. Furthermore, the composition of the
generated aerosol was not dependent on particle size and the only difference between the GF of singly and multiply charged particles is due to the differing magnitude of the curvature effect. For these experiments, the only parameter of interest was the peak diameter of the resultant size distribution, not its breadth. But the distributions were narrow and well-resolved, driven by the TDMA's sheath-to-sample flow ratio of 10:1 (30 vs. 3 l·min$^{-1}$).

CCN activity of aerosol samples from MORA, ACAD, and SPL was analyzed using a DMT-CCNc operated at four fixed supersaturations (nominally 0.2, 0.4, 0.6 and 0.8%) in parallel with the CPC of the TDMA while operating the TDMA as an SMPS. SMPS-CCNc measurements produce two particle size distributions: a conventional distribution based on the response from the CPC and a distribution of the particles that activate in the CCNc. For an internally mixed aerosol, such as the atomizer-generated aerosol measured here, the ratio of the size-dependent concentrations from these two distributions appears as a sigmoidal function that steps from 0 to 1 with increasing particle size. That is, at sizes below the critical activation diameter of the aerosol for the set supersaturation, the CCNc size distribution is zero; above the aerosol's activation diameter the CCN and CCNc distributions are identical. The mid-point of the transition between these regimes ($D_p50\%$) corresponds to the critical dry diameter for CCN activation at the set CCNc supersaturation. These results are typically reported as $D_p50\%(SS_C)$ as in Fig. 2, where $SS_C$ is the aerosol critical supersaturation expressed as a percentage. $D_p50\%(SS_C)$ also translates into $\kappa(SS_C)$, which is comparable to $\kappa(RH)$ (Petters and Kreidenweis, 2013) , as in Fig. 3.

(3)

$$\kappa(SS_C) = \frac{4A^3}{27 \cdot \left[ D_p 50\%(SS_C) \right]^3 \cdot ln^2(SS_C + 1)}$$

This equation is an approximation of $\kappa(SS_C)$, slightly overestimating (~5%) where $\kappa(SS_C)<0.2$. It is retained for its consistency with other studies. *A* is given in Eq. (2) above.

Both instruments were calibrated at the outset of each study and least once more over the course of measurements (~10 days). The critical sizing parameters of the TDMA, the flow meters and high voltage sources, were calibrated directly using a Sensidyne Gilibrator and Fluke multimeter. The RH sensors and CCNc performance were calibrated using the instruments' response to an atomized ammonium sulfate aerosol. The uncertainty associated with TDMA and SMPS-CCNc measurements is primarily driven by uncertainty in RH and supersaturation, respectively. RH was measured using Vaisala HMM-22D RH probes, which have a manufacturer reported accuracy of ±2-3%. The precision of CCNc supersaturation has been reported to range from ±1-5% depending on the stability of ambient conditions and time period (Rose et al., 2008). The uncertainty associated with particle sizing by the TDMA/SMPS instrument, linked to the sheath flow rate measurement (±1%) and high voltage supply (±0.5% for the relevant voltage range), is lower, near ±0.001 μm at 0.070 μm. Additional uncertainty arises from the possibility of irregularly shaped aerosol. Shape factor, the ratio between TDMA-relevant particle mobility diameter and particle volume-equivalent diameter, is used to correct for the depressed mobility of irregularly shaped particles and commonly ranges from 1.02 for ammonium sulfate to 1.08 for cubic sodium chloride to 1.24 for irregularly formed sodium chloride (Wang et al., 2010). Calculations of $\kappa(RH)$ are very sensitive to shape factor at low RH: an uncertainty of ±0.03 in shape factor can translate to >50% uncertainty in $\kappa(RH)$ at low RH but <5% at high RH. No independent recoveries of shape factor, such as SEM micrographs, were made during these studies to constrain this uncertainty. Some confidence can be gained from the continuity between measurements before and after the particles dissolve (and become spherical)—which is detected at low RH for all WSOC samples.

# 3  Results and discussion

WSOC comprised 22–93% of the PM2.5 organic carbon during these studies, with 46–100% recovered in the WSOC isolation process.  Isolated WSOC was characterized by hygroscopicity parameters ($\kappa$) ranging from ~0.05–0.15 calculated from the GF at 90% RH.  In addition to direct contribution to particle water uptake, WSOC appeared to complement the water uptake of inorganic compounds in WSM samples.  Finally, measurements of samples from the GRSM Winter and SPL studies implied that atomizer-generated WSOC particles may assume complex morphologies.

## 3.1 Filter samples and WSOC extracts

As presented in Table 1, WSOC concentration ranged from 0.22 to 1.41 $\mu g \cdot C \cdot m^{-3}$.  The lowest concentration, unsurprisingly, was during the sole winter study (GRSM Winter); otherwise, WSOC varied much more narrowly than reconstructed PM2.5. (PM2.5 was not measured, but Lowenthal et al. (2015) found the sum of measured constituents ($SO_4$, $NO_3$, etc.) reasonably duplicated PM2.5 measurements by co-located IMPROVE network samplers.)  For all studies, the WSOC isolation procedure effectively removed sulfate.  During the four national park studies, the cleaning efficiency was slightly higher, reflecting a slightly different methodology.  For those projects, the samples were reprocessed until the reduction in sulfate reached 99%, likely reducing the fraction of WSOC retained.  The Storm Peak Lab samples were only processed once, to maximize WSOC retention.

## 3.2 Seasonal and locational variation in WSOC hygroscopicity

Figure 2 shows the results of analysis of the samples collected during the five projects.  The pre-desiccated hygroscopic growth curves for each measurement are omitted for clarity, and the scans are normalized with respect to their minimum GF according to the discussion of Sect. 3.6 below.  As reported by Lowenthal et al. (2015), these measurements indicate significant hygroscopic growth of the WSOC aerosol, and a sizable contribution to particulate water.

The measured hygroscopic growth varied little among the samples from each study, but contrasts were apparent between studies.  The least hygroscopic samples were taken at GRSM during the only winter study, likely indicating a smaller relative contribution from highly oxidized, lower molecular weight WSOC.  This is consistent with the seasonal cycle of WSOC characteristics measured by Miyazaki et al. (2012) in a deciduous forest in Japan, which indicated low molecular weight, isoprene-derived WSOC only during summer months.

The differences in hygroscopicity among the samples collected during the summer studies could have various causes. GRSM samples were most hygroscopic.  That site is characterized by being more polluted with much higher particulate mass loading, by deciduous (rather than coniferous) forest, and by being the southernmost site.  From these characteristics, higher levels of isoprene, a more oxidative environment, and greater availability of aqueous aerosol for WSOC partitioning could

be expected, all of which could contribute to WSOC dominated by lower molecular weight, highly oxidized species. Recent studies, e.g., Carlton and Turpin (2013), have emphasized the role of 'anthropogenic' particle water in biogenic SOA formation, especially in the eastern United States.

In Fig. 3, the hygroscopic growth and CCN activity measurements from each study have been recast in terms of $\kappa$, to illustrate the consistency between hydration behavior at high RH and that for supersaturated conditions. The lines are study averages; the error bars indicate the standard deviation of those averaged measurements. In general, there is close agreement between hygroscopic growth at high RH and CCN activity estimates of $\kappa$. Somewhat surprisingly, the largest deviation is lower CCN activity-derived $\kappa$ for the SPL WSM samples: the potential for droplet surface tension depression (leading to

*enhanced* CCN inferred $\kappa$) by surface active organics can be enhanced in mixtures with inorganic ions such as in the WSM (Asa-Awuku et al., 2008).

The shape of these $\kappa$(RH) curves indicates that WSOC behaves somewhat differently than an ideal, aqueous aerosol. Neglecting the influence of surface tension, $\kappa$(RH) for an ideal, fully aqueous aerosol is relatively constant and determined

by relative molecular weights and densities of the solute and solvent (water). We rationalize minor deviations in the shapes of these $\kappa$(RH) curves as non-ideal solution behavior, and abrupt deviations as changes in aerosol phase. In particular, an abrupt decrease in $\kappa$(RH) with decreasing RH in measurements of an initially hydrated aerosol can indicate efflorescence. One caveat to $\kappa$(RH) inferred from hygroscopic growth, especially at low RH, is sensitivity to aerosol shape. When the shape is known, a corrective term—shape factor—is used to adjust the measured mobility diameter to a volume-equivalent

diameter (e.g., cube-shaped, crystalline sodium chloride has the mobility of a spherical particle with 1.08 times its volume) (Wang et al., 2010; Zelenyuk et al., 2006). Here, particles are assumed to be spherical, having a shape factor of 1, with the notable exception discussed in Sect. 3.6 below. Thus, as $\kappa$(RH) for most WSOC samples is fairly constant from 40% RH through CCN activation, it is assumed that the soluble components in these particles are entirely dissolved in that range. However, at low RH, $\kappa$(RH) for the SPL and MORA WSOC samples climbs from near zero to ~0.1, indicating a distinct

deliquescence point (DRH). Neglecting curvature effects, DRH relates to solubility as it is the RH over a solute-saturated solution. This suggests that the solubility of these samples limits the existence of an aqueous phase to RH >30%, *i.e.*, when the activity of the aerosol-phase water is greater than ~0.3. For WSOC collected during the GRSM Winter project, $\kappa$(RH) also increases with increasing RH, but the change is less abrupt. With the remaining WSOC samples there is little change in $\kappa$(RH) over the full RH range, suggesting lower, undetected DRH and higher solubility. The $\kappa$(RH) of WSM samples tends

to increase with RH, especially for samples taken at SPL. Because Fig. 3 shows the behavior of initially hydrated particles, these $\kappa$(RH) curves indicate that some species that make up the WSM gradually leave the aqueous solution as RH decreases. Together with the WSOC measurements, this implies a complementary effect between WSOC and inorganic aerosol material, which is discussed in the next section.

### 3.3 Complementary enhancement of water uptake by mixtures of WSOC and inorganic ions

During two projects, the summers at GRSM and SPL, samples of both ambient WSOC and ambient WSM were isolated and analyzed (only WSOC samples were analyzed for the remaining three projects). In addition to the compounds retained in the WSOC sample, the WSM samples contained all water soluble inorganic and organic material not retained by the WSOC extraction. The hygroscopic growth of these materials indicates that mixtures of WSOC and soluble, inorganic compounds can uptake significantly more water than their individual hygroscopicities would predict. This result is not unanticipated as it is well known that the efflorescence/deliquescence behavior of inorganic salts is modified by the presence of WSOC (Smith et al., 2012; Marcolli et al., 2004; Brooks et al., 2002; Hansson et al., 1998) and that the deliquescence point of an internal mixture of inorganic salts is generally lower than the deliquescence points of the individual compounds (Zardini et al., 2008; Ansari and Pandis, 1999). However, the isolation and analysis of atmospheric WSOC and WSM in the present study provides a unique perspective on these effects in complex, atmospherically representative mixtures.

To evaluate the significance of this complementary effect, the measured WSM hygroscopicity for the SPL and GRSM Summer samples is contrasted with a modeled prediction of WSM hygroscopicity that assumes no interaction between the organic and inorganic fractions—a ZSR-like estimate. Particularly for samples from SPL, this prediction consistently underestimated water uptake by WSM at RH between 30 and 70%, as illustrated in Fig. 4, which suggests that interactions between the inorganic and organic fractions enhance water uptake. As is explored below, this enhancement is linked to the RH at which the inorganic PM dissolves.

### 3.4 Storm Peak Lab: Enhanced water uptake driven by low-DRH WSOC

Enhanced water uptake is most clearly demonstrated in the results from Storm Peak Laboratory in remote northern Colorado. Primarily this is because the SPL study involved more detailed chemical analysis as well as a full set of complementary WSM and WSOC samples. As noted above, the enhancement of water uptake is shown by contrasting the measured WSM hygroscopic growth with a prediction of WSM hygroscopic growth based upon the independent contribution of WSOC and inorganic components assuming no interaction. This prediction is based on the measured WSOC hygroscopicity and the inorganic hygroscopicity estimated from the measured composition.

Modeling the hygroscopic properties from the available SPL inorganic composition was done using the E-AIM thermodynamic model (Clegg and Brimblecombe, 2005; Clegg et al., 1998a, b) and presented several challenges. First, E-AIM does not treat $K^+$, $Mg^{2+}$, and $Ca^{2+}$. However, it does treat multiple solid phases, including hydrates. Critically, it appears to handle the addition of nitrate more realistically than other models. The untreated cations $K^+$ and $Mg^{2+}$ were included in the model input as charge equivalent $Na^+$, while $Ca^{2+}$ was assumed to remove one sulfate ion to form $CaSO_4$ or

gypsum, both of which are relatively insoluble, only expected to dissolve at high RH, and not expected to impact the relevant deliquescence transitions.

A second issue is that that the inorganic anions and cations for these samples do not balance (Hallar et al., 2013). In very similar analysis of WSM and WSOC, Gysel et al. (2004) assumed that a negative ion balance was simply offset with protons. For various reasons, we more closely followed the approach of Hallar et al. (2013) and assumed that the missing cations are ammonium. This approach was chosen largely because the samples all contained significant nitrate which does not typically exist in the aerosol phase with acidic, un-neutralized sulfate (Fountoukis and Nenes, 2007). The cations in the available composition data are insufficient to neutralize the detected sulfate (the average ammonium to sulfate ratio is ~0.93). Moreover, assuming a more acidic aerosol would overestimate the hygroscopic growth. $\kappa$(RH) for ammonium bisulfate, with a molar ratio of ammonium to sulfate of one, is almost 0.8 at 50% RH, much higher than is indicated by the WSM samples. Similarly, each WSM sample exhibits an abrupt increase in $\kappa$(RH) near 80%, which appears to correspond to the deliquescence of the inorganic fraction, but is inconsistent with the acidic mixtures the ammonium to sulfate ratios indicate (DRH of ammonium bisulfate is ~40% RH). The E-AIM results, given as solute mass in and out of solution, were translated into $\kappa$(RH) using the density model of Clegg and Wexler (2011a, b) and are depicted by the dashed lines in Fig 4.

The green curves in Fig. 4 show WSM hygroscopicity as predicted from the independent hygroscopicity of the organic (measured) and inorganic (modeled) fraction of each sample by simply taking the dry solute volume-weighted average of the independent $\kappa$(RH) (Petters and Kreidenweis, 2007). While the dry solute volume of the inorganic fraction can be accurately calculated from the known mass and modeled density, no density information is available for the WSOC. We assume a value of 1.5 g·cm$^{-3}$, but also show that the results are relatively insensitive to this assumption by including the results when assuming 1.3 and 1.7 g·cm$^{-3}$ for contrast. A second minor assumption is that the measured WSOC properties are representative of all the WSOC present in the WSM samples. Other analyses of the SPL samples have indicated that un-extracted WSOC was characterized by relatively lower molecular weight (Hallar et al., 2013; Samburova et al., 2013), which in an ideal mixture would cause greater hygroscopic growth. However, the impact here is expected to be limited; as shown in Table 2, unrecovered WSOC is a minor fraction. Figure 4 shows the result of assuming that the $\kappa$(RH) of the unanalyzed WSOC is 1.5 times that of the analyzed fraction, constraining the likely impact of WSOC not retained by the isolation process. The uncertainty in WSOC density and unanalyzed WSOC is minor and does not appear to account for the enhancement in water uptake by mixed aerosol.

A final note should be made concerning the composition data relied on for this analysis. For Samples 2 and 4, the reported concentrations of inorganic compounds derived from direct analysis of the samples were not consistent with concurrent daily filter-based PM2.5 composition measurements and produced unrealistic predicted WSM hygroscopicity. For this analysis, the relative prevalence of inorganic compounds for those samples was derived from the daily measurements. For Sample 2,

the relative abundance of WSOC was also inferred. In inferring composition, no attempt was made to correct for filter sampling artifacts, such as adsorption of OC, reported by Lowenthal et al. (2009). The potential underestimation is expected to be minor and not impact the overall conclusion.

The measured WSOC hygroscopicity, modeled inorganic component hygroscopicity, and measured and estimated WSM hygroscopicity for each sample from SPL are shown in Fig. 4. As described in Tables 1 and 2, PM2.5 collected during the SPL study was predominantly organic, with an average WSOC to OC ratio of 89%. Each WSOC sample fully deliquesced at RH between 25 and 35%, above which $\kappa(RH)$ remains fairly constant near 0.1. The potential impact of the un-recovered WSOC, assuming it has a $\kappa$ 50% higher than that recovered, is indicated by the smaller markers above each WSOC data

point. Apart from Sample 6, for which the unrecovered fraction was large and $\kappa(RH)$ relatively high, the impact is minor. At low RH, SPL samples exhibit a behavior shared by those from the GRSM Winter study, as well as by the WSOC studied by Gysel et al. (2004): the pre-desiccated measurement-based $\kappa(RH)$ climbs steeply with decreasing RH while that based on the pre-hydrated measurements converge to zero. Sect. 3.6, below, discusses this further, suggesting that it is a measurement artifact related to generated particle morphology. For the purposes of this discussion we assume that, at low RH, the pre-

hydrated measurements are representative of the actual deliquescence properties of the samples.

The inorganic soluble material in SPL WSM samples was dominated by a mixture of sulfate (54% of inorganic mass averaged across samples), nitrate (18%), and ammonium (11%), with minor fractions of Cl$^-$ (<1%), K$^+$ (8%), Na$^+$ (1%), Mg$^{2+}$ (1%), and Ca$^{2+}$ (6%) (Hallar et al., 2013). The hygroscopic growth of the inorganic fraction estimated using the E-AIM

model was fairly consistent from sample to sample, characterized by gradual deliquescence from 45-75% RH. The gradual dissolution predicted by E-AIM is punctuated by the formation of several hydrates and minor salts, but dominated by ammonium sulfate. The model predicts that ammonium sulfate will dissolve gradually below its pure DRH as a solution containing other ions is a more effective solvent (Marcolli et al., 2004). In each sample, the inorganic components are predicted to fully dissolve at RH > 75%.

Measured WSM hygroscopicity of each sample indicates gradual dissolution from 30-80% RH, with a small but abrupt increase around 80%. In Samples 1, 2, and 4, this transition is associated with a small hysteresis loop, which suggests the existence of an ordered, crystalline phase (Martin, 2000). Apart from this small hysteresis loop, there is little difference between the pre-desiccated and pre-hydrated measurements and dissolution and formation of solids appears reversible.

Because the WSOC measurements indicate that it will be dissolved above 30% RH and because WSOC cannot account for $\kappa(RH)$ above ~0.07 ($\kappa(RH)_{WSOC}$ reduced in proportion to WSOC contribution to dry particle volume), the gradual dissolution can be attributed to the inorganic fraction. However, based on the expected deliquescence profile of the inorganic fraction predicted using E-AIM, this dissolution should not take place at all below 45% RH, and should not contribute significantly until ~60% RH. The impact of this shift in the deliquescence RH of the inorganic fraction of the WSM samples is evident in

the contrast between the predicted and measured WSM hygroscopicity. Near 90% RH, where both the predicted and measured WSM $\kappa(RH)$ of most samples have plateaued indicating that all solutes are dissolved, there is reasonably good agreement between them, given the uncertainties involved. The observed difference is most likely due to error in the measured relative abundance of inorganic and organic compounds, which determines the relative weight of the inorganic and

organic hygroscopicities. Errors in WSOC and inorganic hygroscopicity would need to be unrealistically large to close the gap; in Sample 3 for example, $\kappa(RH)$ of the inorganic fraction would need to peak above 0.85. More relevant to this discussion are the errors in the deliquescence profile at lower RH. First, there is some inconsistency in the point at which $\kappa(RH)$ plateaus and all solutes have dissolved. The model captures the profiles of Samples 2 and 3, but predicts full dissolution at too low RH for the other three samples. For all samples, however, the model generally predicts deliquescence

at much higher RH than is evident from the measured WSM profiles. This behavior is consistent with the expectation that DRH is depressed in mixtures (e.g., Marcolli et al., 2004). This trend is even evident near the DRH of WSOC. Because no part of the inorganic fraction is expected to be dissolved below 45% RH, the expected WSM hygroscopicity profile is below even the WSOC curve. Yet $\kappa(RH)$ of the measured WSM is consistently higher than WSOC at low RH. For Sample 2, the measured onset of dissolution of WSM is even lower than that of WSOC.

These results indicate that the hydration behavior of complex, internally mixed aerosol may deviate significantly from that expected based on its major inorganic constituents. The average error of the expected vs. measured WSM hygroscopicity of samples from SPL, shown in Fig. 5, indicates that at low RH the measured $\kappa(RH)$ for WSM is on average twice that expected. This roughly corresponds to a doubling of aerosol water. This behavior is rationalized by depressed DRH of

compounds in mixtures (Smith et al., 2012; Wu et al., 2011; Marcolli et al., 2004; Brooks et al., 2002; Choi and Chan, 2002; Cruz and Pandis, 2000; Hansson et al., 1998). The impact of this behavior is likely widespread as highly soluble, low-DRH WSOC was ubiquitous during these studies. In all SPL WSM samples hysteresis was limited, despite the general expectation that sulfate-nitrate-ammonium aerosols have distinct crystalline and metastable states. Some studies have suggested that mixtures of WSOC and inorganic salts form amorphous, rather than crystalline, phases as they are dried

(Mikhailov et al., 2009). The data from our studies are insufficient to do more than speculate. What is clear is that hysteresis is much less consequential in these mixtures with WSOC and that inorganic compounds can be expected to contribute more to aerosol water in the presence of low-DRH WSOC. The atmospheric impact of this behavior could be significant in dry climates where crystalline aerosol would otherwise form.

### 3.5 GRSM Summer: Mixtures dominated by acidic sulfate

Only two other WSM samples, corresponding to GRSM Summer Samples 4 and 6, were produced and analyzed as part of these studies. Unlike the SPL study, composition had to be inferred from simultaneous daily filter measurements. The GRSM Summer aerosol was also quite distinct from that at SPL. It was dominated by sulfate (9.0 $\mu g \cdot m^{-3}$) that was not fully

neutralized by the available ammonium (1.9 µg·m⁻³). While organic mass was relatively abundant (5.2 µg·m⁻³), only 24% was soluble.

The differences between predicted and measured WSM hygroscopicity for these samples, shown in Fig. 6, may stem from uncertainties in inferred composition. Measured WSM hygroscopicity is much lower than predicted. For both samples closure is possible by assuming the concentration of WSOC is 6.5 times that measured (an average of 7.1 µg·C·m⁻³ instead of 1.1 µg·C·m⁻³). Closure cannot be achieved by assuming realistic changes in the WSOC or inorganic hygroscopicity (i.e., WSOC hygroscopicity is unlikely to be negative). And only a small percentage (~5%) of WSOC hygroscopicity can be attributed to the lower sulfate cleaning efficiency for GRSM Summer samples. Because composition is inferred, it is difficult to speculate whether this error implies a higher than reported ambient WSOC concentration or simply a difference between the samples and daily filter measurements.

As with the SPL samples there is an ion imbalance. The molar ratios of ammonium to sulfate inferred for Samples 4 and 6 are 1.04 and 0.86, respectively. But there is less justification for assuming that ammonium is under-reported. Though sulfate levels have since dropped, GRSM has long been characterized by acidic sulfate aerosol in summer (Lowenthal et al., 2015). Nor was significant nitrate detected, which would have implied neutralized sulfate. The measured Sample 6 WSM hygroscopicity does show behavior consistent with the presence of letovicite, $(NH_4)_3H(SO_4)_2$, which would imply an ammonium to sulfate ratio greater than 1.

Though these results are uncertain, there are several features worth noting, including a potential depression of the deliquescence point of letovicite. Because the dominant fraction of both samples is acidic sulfate, the aerosols are expected to be fully aqueous at all measurement RH. Ammonium bisulfate does exhibit deliquescence (39% DRH) and a crystalline form, but its efflorescence point is below that encountered in the H-TDMA used in this analysis (i.e., <15% RH) (Schlenker and Martin, 2005). Reflecting this, the formation of crystalline ammonium bisulfate is suppressed in the E-AIM estimates shown in Fig. 6. Despite this expectation, Sample 6 exhibits a distinct deliquescence transition near 50% RH. As noted above, this behavior appears to be most consistent with a fraction of crystalline letovicite. Figure 6 illustrates the impact of assuming different ammonium to sulfate ratios (i.e., assuming ammonium was under-reported to differing degrees). Depressed κ(RH) at low RH is due to the formation of crystalline letovicite. As is shown, the expected DRH differs from the measured DRH. E-AIM predicts some depression of DRH of letovicite, as some will dissolve into the surrounding aqueous solution. This is illustrated by the lower DRH of letovicite at lower ammonium-to-sulfate ratios in Fig. 6. There is more aqueous volume in proportion to letovicite and it is entirely consumed at lower RH. Here, the predicted depression was less than measured, suggesting that the presence of aqueous WSOC contributes to the effect.

### 3.6 Irregular particle morphology rationalization for anomalous growth factor at low RH

This section addresses a measurement artifact that has been observed in other similar studies (Boreddy and Kawamura, 2016; Mikhailov et al., 2009; Mikhailov et al., 2004; Gysel et al., 2004). Analysis of samples taken during the GRSM Winter and SPL studies indicates separation between the pre-desiccated and pre-hydrated growth curves at low RH. The measured GF from these projects was also consistently below unity at low RH (as illustrated in Fig. 7, showing the results from GRSM Winter). These results indicate a reduction in particle size as the aerosol is processed between the two DMAs in the TDMA. Though there are several plausible explanations for this behavior, including evaporation of volatile aerosol phase components, the parallels in Gysel et al. (2004) and Mikhailov et al. (2009) support the hypothesis that the size loss is due to the collapse of irregularly shaped particles. Regardless of the cause, in all measurements the pre-desiccated and pre-hydrated profiles eventually overlap and the smallest detected pre-hydrated size is assumed to be the most accurate assessment of dry volume for κ(RH) calculations and for inferring bulk WSOC hygroscopic properties.

The separation between the pre-desiccated and pre-hydrated measurements appears to relate to the aerosol being rehydrated. The aerosol entering the first DMA has been dried to low RH (<20%). The pre-hydrated aerosol is in the collapsed form even during measurements at very low RH because its conditioning begins with rehydration in the humidifying Nafion tube bundle (Fig. 1), while the pre-desiccated particles only collapse when the controlled RH is sufficient for them to substantially dissolve. This link with dissolution is supported by the observance of this behavior only during the GRSM Winter, MORA and SPL projects: As shown in Fig. 3, the WSOC samples from these projects are the only ones that exhibit DRH greater than 30% RH, while the GRSM Summer and ACAD WSOC samples appear fully dissolved at the lowest RH measured. The link to dissolution is consistent with both the collapse of an irregularly shaped aerosol and kinetically limited evaporation. Nor are these rationalizations exclusive as evaporation can produce void fractions within a gel-like aerosol (Mikhailov et al., 2004).

Gysel et al. (2004), Mikhailov et al. (2009), and Boreddy and Kawamura (2016) each report on similar behavior for various WSOC. The report of Gysel et al. is most analogous and attributes this behavior to restructuring. Along with filter extracts of WSOC, they reproduced the phenomenon using Nordic reference humic and fulvic acids and Aldrich humic salts—i.e., substances unlikely to volatilize (Baltensperger et al., 2005). SEM micrographs in Gysel et al. (2004) depict approximately spherical particles; each of the three reports cited above suggests fissures and void fractions account for the apparent reduction in particle density of spray-dried aerosol. Mikhailov et al. (2009) detected similar behavior by oxalic acid aerosol. Though some dicarboxylic acids have been shown to exhibit evaporative losses in TDMAs, Mikhailov et al. (2009) demonstrated the stability of oxalic acid by varying its residence time within the system. Notably, Mikhailov et al. (2009) reported that rapidly drying atomized organic matter in its native, highly charged state could produce high void fraction (40-50%) aerogel-like aerosol. Here, as in Mikhailov et al. (2009), the initial drying of the highly-charged atomizer spray is

rapid and distinct from the drying of singly-charged particles within the TDMA system. The initial drying in this study was not as drastic (~25% vs. ~5% RH) but the similarities in process and result are convincing. These findings parallel other work linking drying rate (Wang et al., 2010) and particle charge (Berkland et al., 2004) to particle morphology. Alternatively, for succinic acid, the solvent used in the spray suspension has been found to dramatically affect morphology

and void fraction (Carver and Snyder, 2012).

In sum, this behavior appears to be a measurement artifact. There is little direct evidence from this study to determine whether it is caused by irregularly shaped particles or evaporative losses, but similar findings by others support the former. For the purposes of this study, the cause is irrelevant and the minimum size reached by the pre-hydrated scan is assumed to

most accurately reflect the amount of aerosol material involved in hygroscopic growth.

## 4 Conclusions

The WSOC in ambient aerosol has been shown to contribute to water uptake through hygroscopic growth and likely through complementary effects with other soluble aerosol components. WSOC was ubiquitous in PM2.5 collected during five month-long studies at various sites and was characterized by hygroscopic growth parameters ($\kappa$) ranging from 0.05 to 0.15.

WSOC samples from GRSM Winter, MORA, and SPL deliquesced near 30% RH, while WSOC samples from GRSM Summer and ACAD did not display deliquescence, but instead were aqueous at all measured RH. No hysteresis was indicated for WSOC samples.

Contrasts between the hygroscopic growth of WSOC and total WSM samples from two of the studies suggest that soluble

components in ambient aerosol can interact to enhance water uptake at atmospherically relevant RH. In particular, highly soluble, low-DRH WSOC can facilitate the gradual dissolution of sulfate-nitrate-ammonium (SNA) at RH below the typical deliquescence point for those substances. The hysteresis behavior of SNA-WSOC mixtures appears to be greatly truncated with mixed SNA-WSOC aerosol hydration instead characterized by gradual, reversible dissolution of SNA as RH increases from 40% to 80%.

Given the atmospheric abundance of internally mixed SNA and WSOC, this study not only indicates that WSOC contributes significantly to aerosol hygroscopicity but that a compartmentalized approach to WSOC and SNA hydration is flawed. The collection, isolation and analysis of WSOC from ambient aerosol provided a new perspective into the hydration behaviors of atmospherically complex mixtures of WSOC, but it is also a broad-brush and imprecise technique. We suggest that its

primary value is in highlighting the complementary effects of WSOC and SNA hydration as a first-order impact on ambient aerosol hydration.

## 5 Data availability

Hygroscopicity and CCN activity data for each sample are included in the article supplement. Data related to the chemical analysis of aerosol samples are reported in Lowenthal et al. (2009, 2015), Hallar et al. (2013), Samburova et al. (2013), and Mazzoleni et al. (2012).

**Acknowledgments.** This work was supported by the EPRI in Palo Alto, CA and by the NSF through collaborative grants AGS-0931431, AGS-0931910, AGS-0931505, and AGS-0931390. Any opinions, findings, and conclusions or recommendations expressed in this material are those of the authors and do not necessarily reflect the views of the National Science Foundation. This research would not have been possible without the cooperation of the National Park Service and

its personnel at Great Smoky Mountains NP, Mount Rainier NP, and Acadia, NP. The Steamboat Ski Resort provided logistical support and in-kind donations. The SPL is an equal opportunity service provider and employer and is a permittee of the Medicine-Bow Routt National Forests.

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

**Tables**

| | GRSM Summer | GRSM Winter | MORA | SPL | ACAD |
|---|---|---|---|---|---|
| Sample period | 7/19/2006 −8/17/2006 | 1/11/2008 −2/9/2008 | 8/1/2009 −8/30/2009 | 6/24/2010 −7/28/2010 | 8/1/2011 −8/30/2011 |
| WSOC/OC | 22% | 21% | 77% | 89% | 93% |
| Sulfate cleaning efficiency | 99.7% | 99.2% | 99.8% | 97.7% | 99.8% |
| Std. Dev. | 0.3% | 0.2% | 0.2% | 2.2% | 0.2% |
| WSOC Recovery | 46% | 100% | 90% | 73% | 60% |
| Std. Dev. | 22% | 17% | 13% | 17% | 70% |
| WSOC ($\mu g \cdot C \cdot m^{-3}$) | 0.64 | 0.22 | 1.41 | 0.72 | 0.78 |
| PM2.5 Reconstructed ($ng \cdot C \cdot m^{-3}$) | 16.2 | 5.3 | 5.0 | 2.3 | 3.1 |

Table 2: Concentration (ng·m$^{-3}$) of Inorganic Ions, Organic Carbon, and Organic Mass in WSM Extracts

| Storm Peak Lab WSM Samples* | | | | | |
|---|---|---|---|---|---|
| Sample Number | 1 | 2 | 3 | 4 | 6 |
| $Na^+$ | 6.5 | 2 | 11 | 8.9 | 6.4 |
| $SO_4^{2-}$ | 544 | 330 (449) | 452 | 77 (488) | 242 |
| $NH_4^+$ | 85 | 58 (80) | 41 | 66 (112) | 33 |
| $NO_3^-$ | 83 | 83 (105) | 137 | 126 (124) | 50 |
| $Cl^-$ | 4.4 | 4.3 | 2.6 | 5.6 | 0.4 |
| $Ca^{2+}$ | 28 | 27 | 27 | 33 | 33 |
| $K^+$ | 42 | 46 | 54 | 44 | 29 |
| $Mg^{2+}$ | 5 | 4.6 | 6 | 7 | 2.2 |
| Total Inorganic Ions | 797 | 555 (718) | 730 | 368 (810) | 396 |
| WSOC (ng·C·m$^{-3}$) | 525 | 1,932 | 601 | 596 | 490 |
| WSOC Recovery (%) | 55 | 87 | 81 | 92 | 52 |
| WSOM (x2.1) | 1,070 | 3,940 (1,390) | 1,226 | 1,220 | 999 |

| GRSM Summer WSM Samples** | | |
|---|---|---|
| Sample Number | 4 | 6 |
| $SO4_4^{2-}$ | 13,200 | 12,000 |
| $NH_4^+$ | 2,580 | 1,950 |
| $NO_3^-$ [†] | --- | --- |
| Total Inorganic Ions | 15,800 | 14,000 |
| WSOC (ng·C·m$^{-3}$) | 1,640 | 533 |
| WSOC Recovery (%)[††] | --- | --- |
| WSOM (x1.8) | 2,950 | 960 |

*Direct analysis of SPL samples was performed and reported by Samburova et al. (2013). Values in parenthesis were inferred from concurrent daily PM2.5 filter analysis reported by Hallar et al. (2013).
**WSM composition was not reported for GRSM Summer. The given values are inferred from concurrent daily PM2.5 filter analysis performed and reported by Lowenthal et al. (2009).
[†]Daily nitrate was not reported; Lowenthal et al. (2009) indicated that 50 ng/m$^3$ was typical during this study.
[††]Daily WSOC Recovery was not reported for these samples; Lowenthal et al. (2009) indicated an average WSOC Recovery of 46% for GRSM Summer.

**Figures**

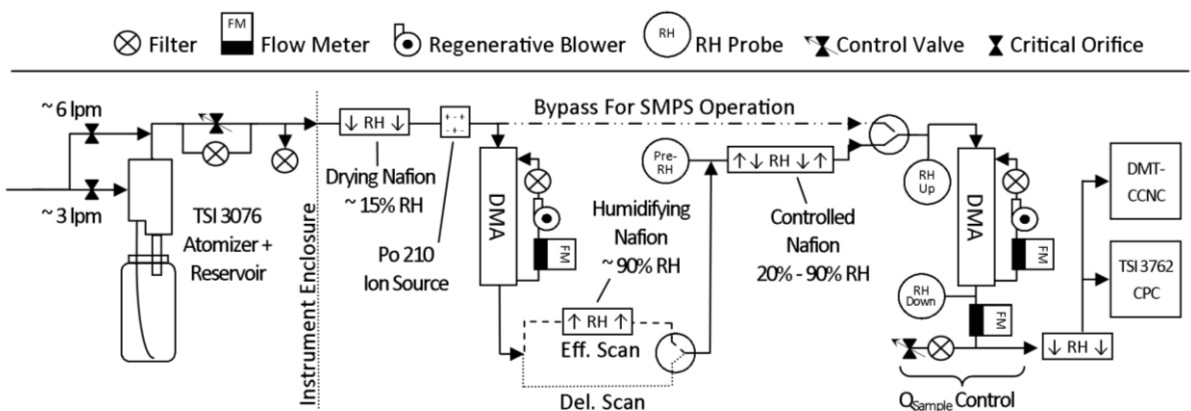

**Figure 1: TDMA and SMPS-CCNc configuration.**

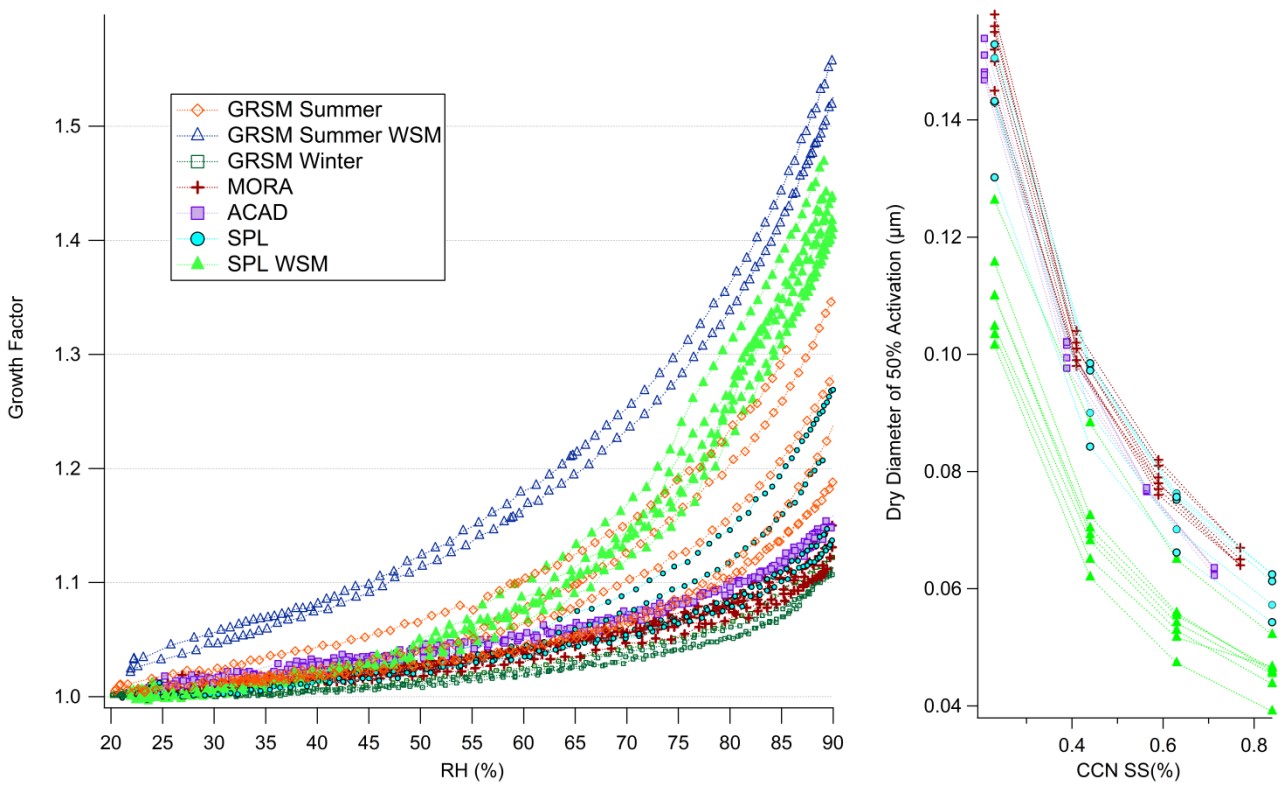

**Figure 2: Hygroscopic growth curves for initially hydrated aerosol samples (designed to detect efflorescence) and CCN activity results. Hygroscopic growth was measured for all samples. CCN measurements were conducted only on MORA, ACAD, and SPL samples (both WSM and WSOC).**

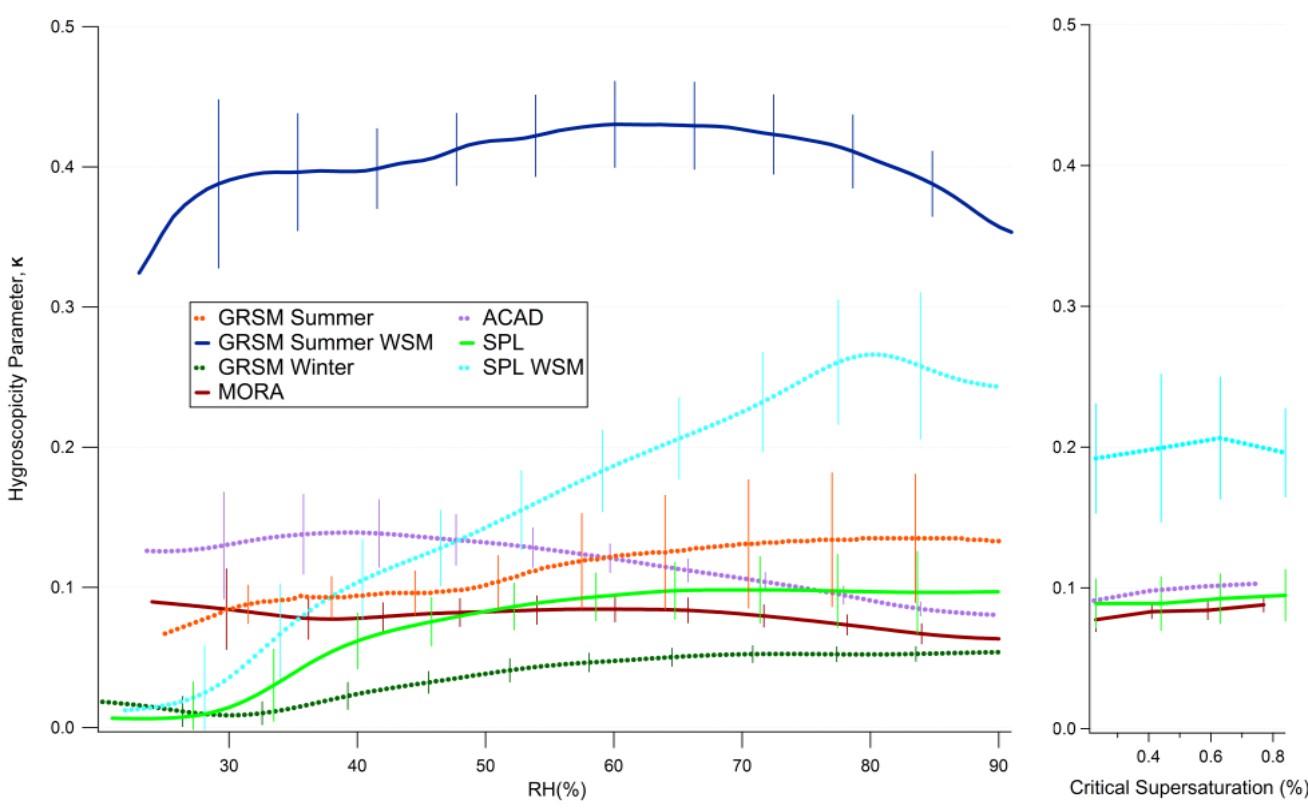

**Figure 3:** Average κ(RH) and κ(SS_c) from all initially hydrated hygroscopicity measurements for all WSOC and WSM samples. Whiskers indicate the standard deviation between samples from each study.

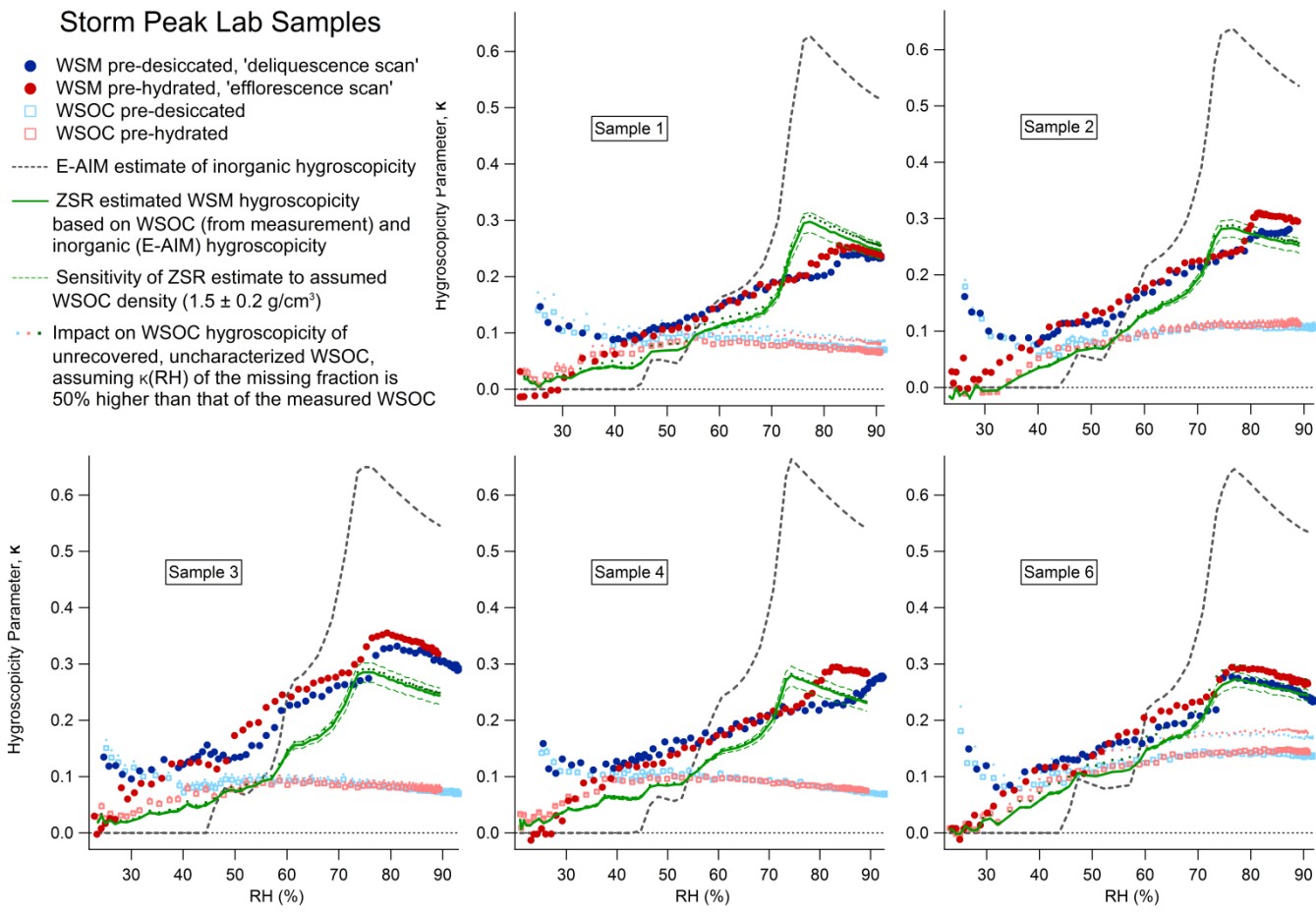

**Figure 4: Comparison of the measured hygroscopicity of SPL WSM samples to expected WSM hygroscopicity assuming no interaction between the organic and inorganic components. WSM hygroscopicity (in green) was estimated from independent assessments of the hygroscopicity of the organic fraction (from WSOC measurements) and of the inorganic fraction (predicted from composition using the E-AIM model). For each sample, WSM hygroscopicity is underestimated below 70% RH, driven by the expectation of the model that the majority of the inorganic fraction will not dissolve until RH reaches ~70%. This suggests that WSOC can significantly depress inorganic DRH, enhancing water uptake below 70% RH.**

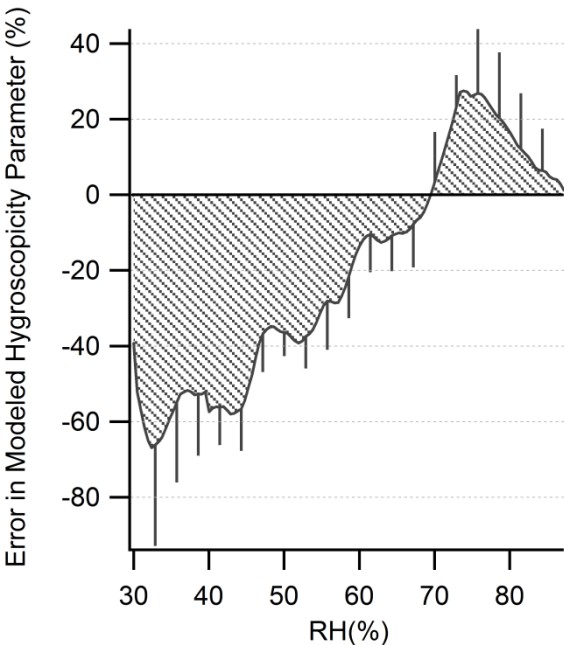

**Figure 5:** Average error in predicted WSM $\kappa(RH)$. Error is defined as $(\kappa(RH)_{predicted} - \kappa(RH)_{measured})/ \kappa(RH)_{measured}$. Whiskers depict the standard deviation of error among the five samples.

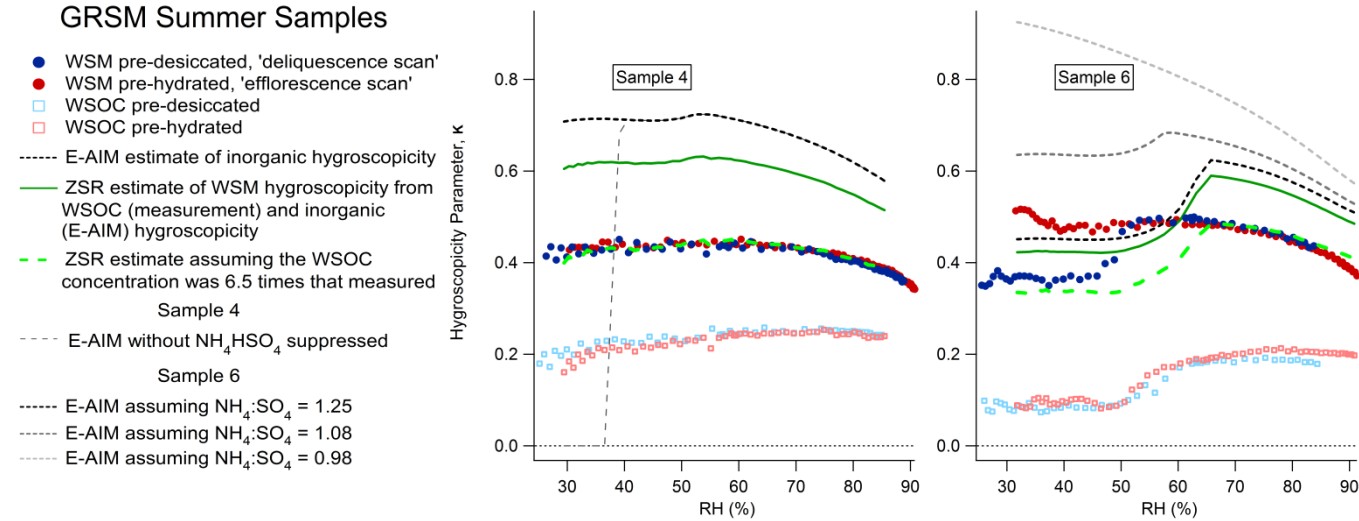

**Figure 6: Contrasting expected and measured WSM hygroscopicity for samples taken during GRSM Summer.**

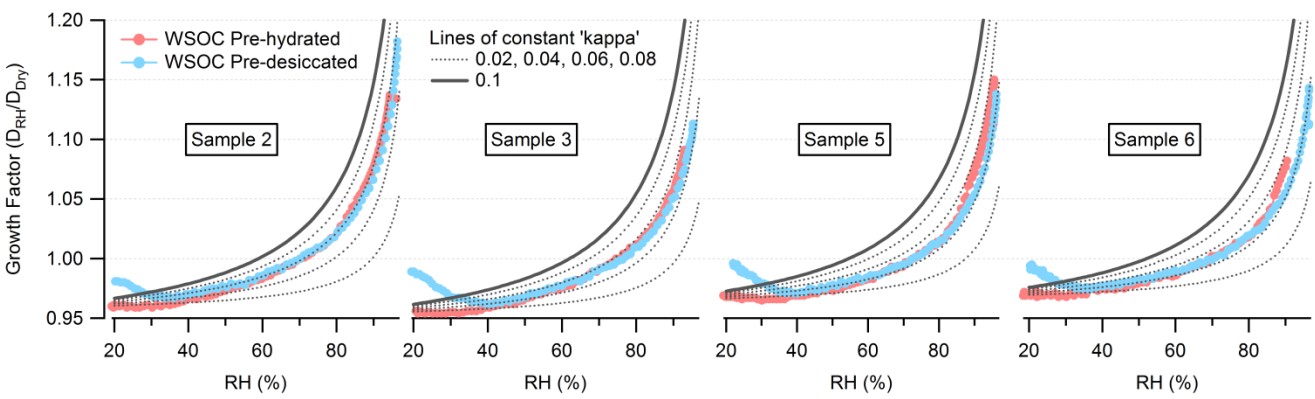

**Figure 7: Hygroscopic growth measurements of WSOC samples from GRSM Winter illustrating below unity growth factor and gradual collapse of pre-desiccated scans at low RH.**

