# Peer review of "Hygroscopic growth of water soluble organic carbon isolated from atmospheric aerosol collected at U.S. national parks and Storm Peak Laboratory"

_Atmospheric Chemistry and Physics, 2016_

## Referee Comment (RC1) · Anonymous Referee #1 · 8 Sep 2016

**General Comments**

This paper presents the results of laboratory experiments performed to characterize the hygroscopicity of water-soluble organic carbon isolated from ambient filter samples collected at several U.S. national parks. Overall, the authors show convincingly that the WSOC makes an important contribution to the hygroscopic properties of the aerosols, even if it is a relatively minor contributor to the PM mass. The findings are significant, and certainly relevant for ACP. The paper is concise, well-organized, and the writing is polished. I highly recommend it for publication after the following items are addressed:

**Specific Comments**

- The biggest weakness (or opportunity for improvement) is that the authors do not extend their analyses to estimate the contribution of WSOC (and inorganic salts) to water uptake as a function of RH for each of the locations/seasons. The major finding in the paper, from the conclusions: "Contrasts between the hygroscopic growth of WSOC and total WSM samples from two of the studies suggest that soluble components in ambient aerosol can interact to enhance water uptake at atmospherically relevant RH." But this is only qualitative, in line with the major discussion surrounding this point in the manuscript. It seems that the authors' current results would enable a more quantitative interpretation of the effects of the WSOC on the hygroscopicity of the ambient particles.

- Two relevant studies are not included, but which should be: The introduction could include a brief discussion of Asa-Awuku et al. (2008), which characterized the CCN activity of WSOC isolated from biomass burning samples. Additionally, Guo et al. (2015) analyzed the contribution of organics to aerosol water in the southeastern U.S. Although their methods were different, the results of Guo et al. (2015) can provide some important context and comparison for the present study, especially the samples from Great Smoky Mountains National Park.

- Should the WSOC concentrations presented throughout the paper have units of $\mu$g-C m$^{-3}$? If not, how have the authors converted from OC to OM?

- I question some of the WSOC concentrations presented in Table 1, especially the samples from GRSM. WSOC contributions of only 4% to the total reconstructed PM$_{2.5}$ mass appear to be unrealistically low.

- I realize that prior papers have presented data from this same study, which the authors cite; however, I think the paper would benefit from added experimental details. Specifically, many of the results compare the pre-hydrated and pre-desiccated scans. Even if this information can be found in another paper, I recommend adding detail to the Methods section to clarify the sequence of the measurements so that the distinction between these measurements is easier to interpret.

- In what ways might the unrecovered WSOC alter the conclusions? The authors have some idea of which types of compounds are likely not recovered – there should be at least a brief discussion of how these compounds could alter the results.

- Finally, in Section 3.6 – what kind of irregular shape do the authors propose? "The hypothesis relies on the desiccated shape produced by drying the atomized aerosol being different than the desiccated shape of the aerosol at low RH in the pre-hydrated measurements." It is not clear how the rate of drying would contribute to this effect? The authors do not believe that they lose WSOC mass in the experimental setup (Section 3.6, line 28). But this at least seems plausible, given recent ambient results (El-Sayed et al., 2016) and laboratory studies (see multiple papers from the De Haan and Turpin groups). Overall, the explanations put forth in this section are shaky, and need further development.

**Technical Corrections**

*none*

**References**

Asa-Awuku, et al.: Investigation of molar volume and surfactant characteristics of water-soluble organic compounds in biomass burning aerosol, *Atmos. Chem. Phys.*, 8, 799-812, 2008.

El-Sayed, et al.: Drying-Induced Evaporation of Secondary Organic Aerosol during Summer, *Environ. Sci. Technol.*, 50, 3626-3633, 2016.

Guo, et al., Fine-particle water and pH in the southeastern United States, *Atmos. Chem. Phys.*, 15, 5211-5228, 2015.

---

## Referee Comment (RC2) · Anonymous Referee #2 · 26 Sep 2016

Review of "Hygroscopic growth of water soluble organic carbon isolated from atmospheric aerosol collected at U.S. national parks and Storm Peak Laboratory" by Taylor et al. (Manuscript Number: acp-2016-715)

**General comments**

This paper presents hygroscopic properties of isolated WSOC and total water-soluble matter (WSM) extracted from PM 2.5 filter samples collected at several U.S. sites. The authors measured the hygroscopic parameters of both WSOC isolated and WSM in the laboratory under different RH conditions. They discuss the relative importance of the isolated WSOC and its interaction with inorganic components in terms of particle hygroscopicity. The present work may provide insights into our understanding on aerosol hydration/dehydration process by organics interacting with inorganics. Overall, however, the manuscript lacks quantitative discussion (see comments below), which makes conclusions rather weak throughout the paper. Moreover, most of the discussion are not convincing because supporting data are not shown in most parts of the paper. The discussion too much relies on the data shown in previous works already published, particularly regarding chemical characterization of the WSM. Although the data presented seems to be valuable, there are a number of important issues that need to be worked out before I recommend its publication in ACP.

**Specific comments**

(1) My major concern is on the exact fraction of WSOC isolated from the WSM compared with "the exact total WSOC" that can be measured without using any XAD columns. In Table 1, although the "WSOC recovery" is shown, it is not clear how the authors calculated/estimated the numbers. It is expected that there might be WSOC components that are not retained either XAD-8 or XAD-4. What is the percentage of these components, and how do these WSOC affect the conclusions? This point should be closely linked to the major conclusion in which the author mentioned the contribution of WSOC to the hygroscopicity of WSM.

(2) How did the authors quantify WSOC? More detailed description on the detection of WSOC is needed including if any syringe filters were used to remove any particulate OC in samples, an instrument used, lower limit of detection, measurement uncertainty, etc. In addition, is the unit of WSOC μg/m3, or μgC/m3? If the authors discuss WSOM

in μg/m3, then how was the mass of WSOC converted to that of WSOM?

(3) Methodology of the hygroscopicity and CCN measurements: The authors should provide more detailed explanations for the settings and calculation with regard to the GF and CCN measurements. Several parameters are not defined in the text, such as water-activity, shape factor, etc. Also, the authors should provide how they derived several important parameters (e.g. κ values), together with some estimation of uncertainty for the GF and CCN measurements. This is important to evaluate the quality of these measurements.

(4) I understand that this study is linked with the previous works in terms of the chemical characterization of aerosols using the same/similar sample sets obtained at the same observational sites. However, almost none of the chemical component in the WSM (besides WSOC) is shown in the manuscript, only referring to the prior papers. The authors should show the amount of each inorganic component (at least SNA) relative to that of WSOC and should add more discussion on that data. Otherwise the discussion is very weak regarding the interaction between WSOC and inorganics, and their contribution to the hygroscopicity.

(5) Table 1: It is not clear how the authors estimated "sulfate cleaning efficiency."

(6) The authors have used the term "highly soluble WSOC" in several parts of the manuscript (e.g., P.10 L.20, P.11, L.7). What is the definition of this "highly soluble WSOC," and what is the difference between this term and "isolated WSOC" which the authors measured?

(7) The authors should add more details on the site descriptions: characteristics of each site (rural, mountain, forest, relative influence of anthropogenic vs. biogenic sources, etc.) should be listed maybe in Table 1 or in an additional table. Also why were those sites selected or appropriate to study the hygroscopicity of WSOC? It is really difficult to understand the major difference among the sites for general readers.

(8) In figures 4-6, the authors
Most of readers may not be interested in the sample number, but interested in the exact difference among the samples in terms of their chemical characteristics.

(9) Section 3.6: Discussion on possible effects of aerosol shape is too speculative. Again, the authors only refer to the previous works and do not show any supporting data to convince readers of their hypothesis. For example, if the authors assume a shape factor other than 1, then how can this support the discussion here?

(10) In Table 1, "SO4" should be "$SO_4^{2-}$" or "sulfate"

---

## Referee Comment (RC3) · Anonymous Referee #3 · 4 Oct 2016

The study reports hygroscopic growth factors, kappa and CCN activity of water-soluble aerosol components in the form of re-aerosolized liquid samples from filters originally collected in National Park locations in the US. The methods, in particular the separation of WSOC from WSM, and the separate analysis of WSOC hygroscopicity, are novel and very interesting to the community. The results show that WSOC has large effects on the hygroscopic behaviors of mixed organic/inorganic aerosols. The paper is well-written and the topic is very relevant. The following points should be addressed before publication:

General comments:

1) The complex experimental setup and measurement program deserve a more detailed description, in particular concerning uncertainties and the exact measurement program. What were the ranges of uncertainty and stability of the various RH settings and flow rates? How were the TDMA data inverted and growth factors determined? What were the uncertainties in the determined activation diameters? The section on TDMA and SMPS-CCNC operation would also profit from a better links between the text and Figure 1: What were the setpoints of size and RH at which points in the setup and in time for which measurement series?

2) More detail should also be provided on the handling and chemical analysis of the filter samples. How many filters, and which days were combined for each re-aerosolized sample? How might aerosol properties have changed over the course of the days combined into each sample? Also, it appears that some of the filter samples were taken as long as ten years ago - when were they extracted, and when were the laboratory measurements done? Can it be ensured that there are no artefacts from storage due to evaporation or other processes? In the results, fractions of WSOC and OC are reported, as well as WSOC retention percentages. How was this assessed, was there an independent WSOC measurement? How (and when) was OC measured? It would also be much more reader-friendly to present the inorganic composition of the combined samples in this paper, rather than (more or less explicitly) referring to earlier studies (do they feature the same combination of filters to larger samples?).

3) The discussion on the enhancement of hygroscopic growth through WSOC could, and should, be more quantitative. "Enhancements" would be more obvious if measured values were compared to reference values - either of measured hygroscopic properties of the inorganic components (not just the total WSM) or of theoretical values/theoretical growth curves of inorganic salts in the Figures. See more specific comments below.

4) The discussion of impact of particle shape needs substantiation. It seems a stretch

to invoke two different mechanisms of particle shape changes as an explanation for both the deliquescence and efflorescence branch of the growth curves. The collapse of particle stuctures upon hydration is a known effect for aerosol types such as fresh combustion agglomerates, but it is a far less obvious thought for a re-aerosolized WSOC sample. How do the authors know that the atomized aerosol is "irregularly shaped"? A discussion of the growth factor uncertainties deriving from the experimental setup and from GF determination from the DMA2 size distribution should be given. Also, why are the GF<1 not showing in the GRSM GF curve in Figure 2?

Detailed comments:

p.4, Section 1.1: Please give more details on those locations and sampling sites.

p.6, line 20: Why was this exact dry diameter chosen?

p.6, line 24: How often is "periodically"?

p.7, lines 14-16: This should be described earlier, along with more details on the measurement methods and results of the inorganic compounds.

p.7, line 23: Please specify here: which one is the winter study? It is not reader-friendly to have him/her leaf back to the study description, find the abbreviation that refers to the winter study, and then re-locate that abbreviation in Figure 2.

Figure 2: a) dark blue and dark green, as well as light blue and light green are hard to distinguish. How about a different symbol but the same color for WSM and WSOC of the same location? It would also be nice to call "GRSM I"/"GRSM II" "GRSM summer" and "GRSM winter", to spare the reader repeated leafing back for which one is which. b) This presentation of activation diameters should be commented on in more detail in the text. Alternatively, it could be dropped.

p.8, line 19: what kind of change classifies as "minor", what as "large"? Please expand (or add a reference) on how a "large" change in kappa indicates a phase change.

p.8, line 30-31: In Figure 3, GRSM I starts at around 0.3 -I would not call this "near zero".

p.9, line 8: "complementary enhancement" as compared to what? The inorganic components alone? (Below or above efflorescence?) For this, a reference hygroscopicity value (measured or calculated) of the isolated inorganic components should be given. What is currently shown in Figure 3 is just that total WSM has a higher kappa than the WSOM, which is not surprising. The paper would improve substantially if this analysis could be more quantitative.

p.9, line 6-7: These chemical compositions should be shown in a figure.

p.9, line 20: Where and how is this shown?

p.9, line 26: Where is this argument going?

---

## Author Response (AR1)

Dear Editor and Referees:

We would like to thank our reviewers for taking time to evaluate this manuscript and provide thoughtful comments.  The primary criticism, that our paper lacked sufficient quantitative support for its claims, was well made and led to substantial improvement in our analysis and manuscript. We have addressed this criticism and all other comments in the attached supplement.

Our response addresses each reviewer in turn.  Referee comments are in **bold,** followed by our response in plain text.  Changes to the manuscript are given in *italics.*  Specific changes are in red.  Many of the comments are shared by each referee.  For brevity, we only address these points fully when they are first raised, but provide brief summaries and references in response to subsequent similar comments.

**Referee Comment 1:**

**This paper presents the results of laboratory experiments performed to characterize the hygroscopicity of water-soluble organic carbon isolated from ambient filter samples collected at several U.S. national parks. Overall, the authors show convincingly that the WSOC makes an important contribution to the hygroscopic properties of the aerosols, even if it is a relatively minor contributor to the PM mass. The findings are significant and certainly relevant for ACP. The paper is concise, well-organized, and the writing is polished. I highly recommend it for publication after the following items are addressed:**

Thank you for taking time to review this work and for affirming the value of the results. We believe that we have answered each of the following comments leading to significant improvements of the manuscript.

**Comment 1.1**

**The biggest weakness (or opportunity for improvement) is that the authors do not extend their analyses to estimate the contribution of WSOC (and inorganic salts) to water uptake as a function of RH for each of the locations/seasons. The major finding in the paper, from the conclusions: "Contrasts between the hygroscopic growth of WSOC and total WSM samples from two of the studies suggest that soluble components in ambient aerosol can interact to enhance water uptake at atmospherically relevant RH." But this is only qualitative, in line with the major discussion surrounding this point in the manuscript. It seems that the authors' current results would enable a more quantitative interpretation of the effects of the WSOC on the hygroscopicity of the ambient particles.**

The criticism that the paper is too qualitative is well taken, and repeated by all referees. Each reviewer primarily criticized the lack of quantification of the tendency of WSOC to enhance the water uptake by the inorganic fraction. Our response and description of the steps taken to rectify this issue are given here and will be referenced in answers below.

Referee 1 makes two suggestions here: a) that the estimated contribution of WSOC and inorganic salts to water uptake be extended to all locations and seasons, and b) that the impact of WSOC on the hygroscopic growth of ambient particles be quantified.

*In response to the first suggestion*, we acknowledge that in isolation, WSOC hygroscopicity could be extrapolated based on atmospheric conditions to estimate seasonal contribution to particle water. However, the focus of this manuscript is on the enhanced uptake of water due to interactions between WSOC and inorganic ions. This effect is more difficult to extrapolate as there are unfortunate limitations in our dataset. Mainly, extracts containing all water soluble aerosol constituents (Water Soluble Materials or WSM) were not prepared for most seasons and locations. These studies were narrowly designed to demonstrate WSOC hygroscopicity; the findings presented here are somewhat incidental. Likewise, only the Storm Peak Lab study, part of a separate project, provided detailed chemical speciation. Thus, it is difficult to do much more than speculate about the broad contribution of WSOC to hygroscopic growth during all

seasons and locations. But we have made significant changes to the manuscript attempting to quantify these effects and indicate why, based on our limited data, they might be widespread.

> The manuscript was significantly expanded to quantify of the enhancement of water uptake due to interactions between organics as discussed in the following response. Sections 3.3-3.5 were largely rewritten to address this point. The effect on particle water at SPL is described in Fig. 5 and discussed on page 15, lines 20-26. The potential to extrapolate these effects is based on the ubiquitous presence of low-DRH WSOC during these studies, which is emphasized frequently (e.g. p.15, line 26; p.18, line 15). However, extrapolating would require knowledge or estimation of WSM properties (only available for SPL) and is beyond the scope of this work.
* * *
*The second suggestion* by Referee 1—also made by Referees 2 and 3—requests that we quantify the impact of WSOC on aerosol water uptake by the full mixture of water soluble aerosol material (WSM). The point is well taken and we believe the changes made have improved the manuscript.

The referees' criticism focuses on our qualitative assertion that, because WSOC deliquesces at lower RH than many inorganic aerosol constituents (or is aqueous throughout the measure RH range), it likely allows inorganic components to enter solution at lower RH than they would in isolation. The referees' comments ask that we further quantify this enhancement and more precisely isolate the effect of WSOC.

In response, we have adopted the referees' suggested approach. To quantify and isolate the enhanced water uptake of WSOC and inorganics in mixtures, we have contrasted the measured WSM hygroscopic growth with a prediction of WSM growth that assumes no interaction between the organic and inorganic fraction—essentially a ZSR-like estimate. This allows us to attribute the contrast between observed and expected WSM hygroscopicity to the interaction of WSOM and inorganics.

> To address this comment, we have re-written Sections 3.3-3.5 (pages 12-16) to incorporate a more sophisticated treatment of this effect. The extended response provided here (below) is largely duplicative of the reworked portion of the manuscript.

Our approach is similar to that employed by Gysel et al. (2004). The 'expected' WSM hygroscopic growth is estimated from the hygroscopic properties of the inorganic (modeled) and organic (WSOM, measured) water soluble compounds, i.e., the two components of WSM. The inorganic contribution is estimated using the E-AIM model of Clegg and Wexler (see list of E-Aim publications below). WSOM hygroscopicity was directly measured using the methods described in the paper. The expected hygroscopic growth is predicted assuming no interaction—which is a straightforward volume (of the solutes) weighted average of the hygroscopicity parameter.

There are a number of key assumptions in each of these three steps:  modeling inorganic hygroscopicity, applying the measured hygroscopicity of WSOM, and predicting their combined hygroscopicity.

*Modeling Inorganic Hygroscopic Growth*

We evaluated two hygroscopic growth models—E-AIM and ISORROPIA II—for this task and decided the E-AIM model was more appropriate.  Though the inorganic composition was analyzed for the SPL study, there were several difficulties in producing what we believe to be supportable and reasonable estimates of hygroscopic growth.  First, the composition data indicates both that there are insufficient cations to neutralize the sulfate *and* a sizable portion of nitrate.  Second, several of the cations detected are not treated by the E-AIM model.

The first issue arises because the presence of nitrate requires that sulfate in the samples be neutralized. (Fountoukis, C. and A. Nenes, 2007).  However, the cations measured do not adequately account for neutralized sulfate.  We resolve this issue by assuming that the ammonium was under-measured and model the hygroscopic growth by balancing the measured anions with additional ammonium (rather than protons).

This approach—assuming that the sulfate was in fact neutralized by ammonium—is supported by two strong arguments and two weaker arguments.   First, if the sulfate was not neutralized and following the approach of Gysel et al. (2004) we assumed that the missing cations were offset with $H^+$, the resulting estimation of WSM hygroscopicity is clearly unrealistic.  Often this would imply a sizable fraction of ammonium bisulfate (k(50% RH) = ~0.8), drastically over predicting the WSM hygroscopicity.  While our use of this model *is* intended to demonstrate a similar error (underestimation), underestimation is consistent with expected behavior: components with low deliquescence points are well known to enhance water uptake by less soluble aerosol components.  Second is the consistent presence of nitrate, which does not partition to particles containing acidic sulfate.  Third, it is supported by the difficulty of measuring ammonium.  Finally, it is supported by the appearance of hysteresis and deliquescence at high RH (~80%) in each of the measurements, which is consistent with a major fraction of ammonium sulfate.  Unfortunately, this all introduces some uncertainty about the actual characteristics of the inorganic fraction and undermines the ultimate conclusion that highly soluble organic matter enhances water uptake.  In the end, we increased the amount of ammonium input to the model to fully neutralize the sulfate.

We deal with the second issue, the presence of cations not treated by E-AIM ($Ca^{2+}$, $K^+$, $Mg^{2+}$) by either substituting $Na^+$ ($K^+$, $Mg^{2+}$), or suppressing $SO_4^{2-}$ ($Ca^{2+}$, reflecting the much lower solubility of gypsum).  Though this is not an exact approach, these are minor constituents.

Finally, we translate the model results into k(RH).  k(RH) is primarily a volume based property and we use the density calculator hosted on the E-AIM web site to translate the aqueous solutions into volumes.

> *Changes in the manuscript:  We have included the estimated hygroscopic growth of the inorganic species in Figures 4 and 6, which depict the hygroscopic growth of the SPL and*

*GRSM Summer samples.  We have also added a similar discussion to the text, describing our approach to modeling these compounds.*

*Applying the Measured Hygroscopicity of WSOC*

The studies reported in this paper all involved measuring the hygroscopicity of samples of WSOM isolated from ambient aerosol using a TDMA.  As we will return to in our response to later comments, there are two main gaps between these measurements and the hygroscopic properties of the WSOM in the full WSM (and in ambient aerosol).

First, TDMA measurements are reflective of mobility diameter of particles, which does not necessarily translate accurately to volume.  The hygroscopic growth parameter, k(RH), is derived from changes in particle volume and is thus similarly uncertain.  While we believe that there is good reason to suspect that these particles are nearly spherical (e.g., Gysel et al., 2004; Mikhailov et al., 2009; Boreddy and Kawamura, 2016) there is simply no definitive evidence to evaluate this contention.  We expand on this limitation in the paper and below.

Second, while the WSOM extracts typically represented a large majority of the total WSOM in the WSOC, a significant fraction was not captured.  It is known from analyses reported in other papers that the missing fraction is expected to include smaller organic molecules—and is therefore likely to enhance the hygroscopicity of the WSM to a greater degree than the WSOM extracts would indicate.

*Changes in the manuscript:   To attempt to capture the impact of this missing fraction we have added error bars to both the WSOM and WSOM+INORGANIC estimate indicating the impact if the missing fraction was characterized by k(RH) 50% higher than the WSOM extract.  In addition, we have added a description of this impact to the text.*

*Estimating WSM hygroscopic growth from modeled inorganic and measured WSOM hygroscopic growth*

Once we have a k(RH) curve representative of the inorganic and organic fractions, it is straightforward to estimate k(RH)$_{mixed}$, assuming no interaction.  It is simply the volume weighted average of k(RH)'s of the individual solutes. (Petters, M. D. and S. M. Kreidenweis, 2007)  Because the chemical analysis of the WSM only provides component mass, it is necessary to assume the density of WSOM (the density of the inorganic compounds involved are well known).  We assume a density of 1.5 g/cm$^3$ for WSOM, but include error bars to indicate the wide uncertainty this introduces.

*Changes in the manuscript:  Beyond the section describing our approach, we include the results of our modeling effort in Figures 4 and 6, to contrast that expected hygroscopic growth with the measured hygroscopic growth of WSM.*

Papers Cited in this response:

Gysel, M., E. Weingartner, et al. (2004). "Hygroscopic properties of water-soluble matter and humic-like organics in atmospheric fine aerosol." Atmos. Chem. Phys. 4(1): 35-50.

Boreddy, S. K. R. and K. Kawamura (2016). "Hygroscopic growth of water-soluble matter extracted from remote marine aerosols over the western North Pacific: Influence of pollutants transported from East Asia." Science of The Total Environment 557–558: 285-295.

Mikhailov, E., S. Vlasenko, et al. (2009). "Amorphous and crystalline aerosol particles interacting with water vapor: conceptual framework and experimental evidence for restructuring, phase transitions and kinetic limitations." Atmos. Chem. Phys. 9(24): 9491-9522.

Petters, M. D. and S. M. Kreidenweis (2007). "A single parameter representation of hygroscopic growth and cloud condensation nucleus activity." Atmos. Chem. Phys. 7(8): 1961-1971.

Fountoukis, C. and A. Nenes (2007). "ISORROPIA II: a computationally efficient thermodynamic equilibrium model for $K^+$–$Ca^{2+}$–$Mg^{2+}$–$NH_4^+$–$Na^+$–$SO_4^{2-}$–$NO_3^-$–$Cl^-$–$H_2O$ aerosols." Atmos. Chem. Phys. 7(17): 4639-4659.

The E-Aim Papers

Carslaw, K. S., S. L. Clegg, et al. (1995). "null." The Journal of Physical Chemistry 99(29): 11557-11574.

Clegg, S. L. and P. Brimblecombe (2005). "Comment on the "Thermodynamic Dissociation Constant of the Bisulfate Ion from Raman and Ion Interaction Modeling Studies of Aqueous Sulfuric Acid at Low Temperatures"." The Journal of Physical Chemistry A 109(11): 2703-2706.

Clegg, S. L., P. Brimblecombe, et al. (1998). "Thermodynamic Model of the System H+−NH4+−Na+−SO42-−NO3-−Cl-−H2O at 298.15 K." The Journal of Physical Chemistry A 102(12): 2155-2171.

Clegg, S. L., P. Brimblecombe, et al. (1998). "Thermodynamic Model of the System H+−NH4+−SO42-−NO3-−H2O at Tropospheric Temperatures." The Journal of Physical Chemistry A 102(12): 2137-2154.

Massucci, M., S. L. Clegg, et al. (1999). "Equilibrium Partial Pressures, Thermodynamic Properties of Aqueous and Solid Phases, and Cl2 Production from Aqueous HCl and HNO3 and Their Mixtures." The Journal of Physical Chemistry A 103(21): 4209-4226.

**Comment 1.2:**

**Two relevant studies are not included, but which should be: The introduction could include a brief discussion of Asa -Awuku et al. (2008), which characterized the CCN activity of WSOC isolated from biomass burning samples. Additionally, Guo et al. (2015)**

analyzed the contribution of organics to aerosol water in the southeastern U.S. Although their methods were different, the results of Guo et al. (2015) can provide some important context and comparison for the present study, especially the samples from Great Smoky Mountains National Park.

*Changes in manuscript: The Asa-Awuku et al. study is extremely relevant, relies on similar use of solid phase extraction, and will be included in the introduction. Similarly, we have added Guo et al. to our discussion of WSOC prevalence and contribution to particle water.*

Asa-Awuku et al. (2008) is discussed on page 2, lines 2-8 and page 11, lines 9-11.

Guo et al. (2015) is discussed with on page 3, lines 24-25.

**Comment 1.3:**

**Should the WSOC concentrations presented throughout the paper have units of µg -C m -3 ? If not, how have the authors converted from OC to OM?**

*Changes in manuscript: Referee 1 is correct, the units of WSOC would be most clearly conveyed as µg-C·m$^{-3}$. We have revised the manuscript accordingly.*

This change has been incorporated and µgC/m$^3$ is used where possible for the sake of clarity.

**Comment 1.4:**

**I question some of the WSOC concentrations presented in Table 1, especially the samples from GRSM. WSOC contributions of only 4% to the total reconstructed PM 2.5 mass appear to be unrealistically low.**

This is a reasonable question, but these results are consistent with the co-located IMPROVE site measurements for the same period. Sulfate was typically reported in the 15 µg/m$^3$ range, and the total OC values were around ~3 µg-C/m$^3$. Our results indicated that WSOC was ~22% of OC, which is lower than typical.

**Comment 1.5**

**I realize that prior papers have presented data from this same study, which the authors cite; however, I think the paper would benefit from added experimental details. Specifically, many of the results compare the pre-hydrated and pre-desiccated scans. Even if this information can be found in another paper, I recommend adding detail to the Methods section to clarify the sequence of the measurements so that the distinction between them is evident.**

*Changes in manuscript: In response to this comment and similar comments below, we have added more details to the methods section including descriptions of the pre-hydrated and pre-desiccated scans.*

The methods section has been expanded significantly.

P.7, lines 21-32 describes the pre-hydrated and pre-desiccated scans in much greater detail, linking their operation to Fig. 1.

**Comment 1.6**

**In what ways might the unrecovered WSOC alter the conclusions? The authors have some idea of which types of compounds are likely not recovered – there should be at least a brief discussion of how these compounds could alter the results.**

In response to this (and similar comments), we have expanded our discussion and analysis of the contribution of un-recovered WSOC. The fraction of WSOC in the WSM that was not isolated is likely comprised of smaller organic molecules; k(RH) measured of the isolated WSOC will likely underestimate the actual contribution of WSOC to WSM (and ultimately ambient aerosol) hygroscopicity.

*Changes in manuscript: To account for this effect, we have included error bars in Figures 4 and 5 that illustrate, given the amount of WSOC that was not recovered, how much enhancement in total WSOC hygroscopicity we might expect. We have also added a description of this source of error to the discussion.*

The effect of unrecovered WSOC is now discussed on page 13, lines 24-31 and depicted in Fig. 4.

**Comment 1.7**

**Finally, in Section 3.6 – what kind of irregular shape do the authors propose? "The hypothesis relies on the desiccated shape produced by drying the atomized aerosol being different than the desiccated shape of the aerosol at low RH in the pre-hydrated measurements." It is not clear how the rate of drying would contribute to this effect? The authors do not believe that they lose WSOC mass in the experimental setup (Section 3.6, line 28) . But this at least seems plausible, given recent ambient results (El- Sayed et al., 2016) and laboratory studies (see multiple papers from the De Haan and Turpin groups). Overall, the explanations put forth in this section are shaky, and need further development .**

This point is well received—we do not have any measurements or data that would provide a definitive explanation for this behavior. The discussion was originally included because as earlier, similar studies have reported similar behavior (e.g., Gysel et al., 2004) we felt it was worthwhile to indicate that we had noticed the same—and give a few off-the-cuff rationalizations. We also included this section to explain how we selected a baseline for our growth factor measurements.

We have done further research on the topic. Several other groups have detected the same behavior and we have reworked our discussion to draw more heavily on those studies (rather than guesswork). In particular, Gysel et al. (2004) look more systematically at this behavior and described it as restructuring. They used SEM to look at the particle shapes, which were spherical. They also recreated the behavior with of well-known non-volatile compounds: with nordic reference humic and fulvic acids and salts. Boreddy and Kawamura (2016) report on a similar behavior, and mainly adopted the rationalization of Gysel et al.—which hinged on rates of evaporation leading to internal cracks and voids.

A similar behavior was also noticed by Mikhailov et al. (2009) for oxalic acid and levoglucosan. Their investigation found that spray drying could produce particles with high void fractions that collapse through humidification/drying cycles.

Still, it is possible that this is an artifact of evaporation, and we will note that in our discussion.

Section 3.6 (pp.17,18) has been largely rewritten in response to this and similar comments, focusing on similar reports by Gysel et al. and others.

Papers cited in response

Gysel, M., E. Weingartner, et al. (2004). "Hygroscopic properties of water-soluble matter and humic-like organics in atmospheric fine aerosol." Atmos. Chem. Phys. 4(1): 35-50.

Boreddy, S. K. R. and K. Kawamura (2016). "Hygroscopic growth of water-soluble matter extracted from remote marine aerosols over the western North Pacific: Influence of pollutants transported from East Asia." Science of The Total Environment 557–558: 285-295.

Mikhailov, E., S. Vlasenko, et al. (2009). "Amorphous and crystalline aerosol particles interacting with water vapor: conceptual framework and experimental evidence for restructuring, phase transitions and kinetic limitations." Atmos. Chem. Phys. 9(24): 9491-9522.

**Technical Corrections**

 **none**

**Referee Comment 2:**

General comments

**This paper presents hygroscopic properties of isolated WSOC and total water-soluble matter (WSM) extracted from PM 2.5 filter samples collected at several U.S. sites. The authors measured the hygroscopic parameters of both WSOC isolated and WSM in the laboratory under different RH conditions. They discuss the relative importance of the isolated WSOC and its interaction with inorganic components in terms of particle hygroscopicity. The present work may provide insights into our understanding on aerosol hydration/dehydration process by organics interacting with inorganics. Overall, however, the manuscript lacks quantitative discussion (see comments below), which makes conclusions rather weak throughout the paper. Moreover, most of the discussion are not convincing because supporting data are not shown in most parts of the paper. The discussion too much relies on the data shown in previous works already published, particularly regarding chemical characterization of the WSM. Although the data presented seems to be valuable, there are a number of important issues that need to be worked out before I recommend its publication in ACP.**

We thank Referee 2 for thoughtfully evaluating this manuscript. We hope that we have addressed the major concern, the lack of quantitative evaluation. As we describe in our response to Referee Comment 1.1, we have expanded our analysis to isolate and quantify the contribution of WSOC to WSM hygroscopic growth. We have also sought to include more of the chemical analysis results (from earlier publications) on which we have based this quantification.

**Specific comments**

**(1) My major concern is on the exact fraction of WSOC isolated from the WSM compared with "the exact total WSOC" that can be measured without using any XAD columns. In Table 1, although the "WSOC recovery" is shown, it is not clear how the authors calculated/estimated the numbers. It is expected that there might be WSOC components that are not retained either XAD-8 or XAD-4. What is the percentage of these components, and how do these WSOC affect the conclusions? This point should be closely linked to the major conclusion in which the author mentioned the contribution of WSOC to the hygroscopicity of WSM.**

In response to Referee 2's methodological questions, we have expanded our description of the TOC analysis methods employed to determine the OC content of the WSM and WSOC, as well as the details of the "WSOC recovery calculation."

The second point made by Referee 2 is appreciated: we have expanded our analysis of the impact of the WSOC that was not retained in the resin-isolated WSOC samples. Our approach is covered in more detail in our response to a similar comment by Referee 1—in responses 1.1 and 1.6. As this comment notes, the discussion is tightly tied to our quantification of the contribution of WSOC to WSM hygroscopicity. In brief, we illustrate and discuss the sensitivity

of our conclusions to the possibility that the missing WSOC has different characteristics than the WSOC analyzed by TDMA.

*Changes in manuscript: We have elaborated on the underlying chemical analysis from earlier projects, providing more details and more data. We have also greatly expanded our evaluation of WSOC's contribution to WSM hygroscopicity, including the possible impact of WSOC not isolated and analyzed.*

The methods section, at page 6, lines 11-25, has been expanded to include descriptions of the chemical analysis of the filters and extracts during these studies. In particular, the methods of analyzing WSOC are described in lines 15-23, including the determination of WSOC recovery.

The effect of unrecovered WSOC is now discussed on page 13, lines 24-31 and depicted in Fig. 4.

**(2) How did the authors quantify WSOC? More detailed description on the detection of WSOC is needed including if any syringe filters were used to remove any particulate OC in samples, an instrument used, lower limit of detection, measurement uncertainty, etc. In addition, is the unit of WSOC μg/m3, or μgC/m3? If the authors discuss WSOM in μg/m3, then how was the mass of WSOC converted to that of WSOM?**

Beyond answering these reasonable questions, we have generally expanded the discussion of the production of WSOC extracts. We appreciate the criticism that we did not adequately incorporate findings reported in earlier studies and have attempted to include more details.

*Changes in manuscript: The specific questions about WSOC extraction and chemical analysis are now answered in the methods section. The distinction between WSOM and WSOC is clarified by incorporating units of μgC/m$^3$. We have also indicated the conversion from OC to OM used for the National Park studies, 1.8, and for the Storm Peak Lab Study, 2.1.*

A detailed discussion of WSOC chemical analysis was added on page 6, in lines 15-23, which describes the instrumentation and process of quantifying WSOC, including measurement uncertainty.

The description of extract preparation has also been expanded, beginning on page 5, line 24, and reports the use of a PTFE membrane filter.

In response to this and similar comments, WSOC is now consistently reported in μgC/m$^3$. The conversion between WSOC and WSOM is reported on page 6, line 21.

**(3) Methodology of the hygroscopicity and CCN measurements: The authors should provide more detailed explanations for the settings and calculation with regard to the GF and CCN measurements. Several parameters are not defined in the text, such as water-activity, shape factor, etc. Also, the authors should provide how they derived several important parameters (e .g. κ values), together with some estimation of uncertainty for**

**the GF and CCN measurements. This is important to evaluate the quality of these measurements.**

*Changes in manuscript: In the methodology section, we have expanded our description of the hygroscopicity and CCN activity measurements. We provide a rough estimate of uncertainty for both the TDMA and CCNc instruments. Also, in keeping with our extended analysis of the relative contributions of WSOC and inorganic ions to WSM hygroscopicity, included more details of the calculations involved.*

The description of hygroscopicity and CCN measurements has been greatly expanded in Section 2.5, beginning on page 7, in response to this and similar comments. The instrument settings (p.7, lines 19-32; p.8, line 28-32) and data processing routines (p.8, lines 1-8, 21-26) are discussed in greater depth, as are the measurement uncertainties (p.9, lines 12-28).

Water activity has been largely eliminated from the manuscript, while shape factor has been elaborated on page 9, lines 21-28 and page 11, lines 15-18.

The equations used for calculating κ are now included on page 8, line 10 and page 9, line 10.

**(4) I understand that this study is linked with the previous works in terms of the chemical characterization of aerosols using the same/similar sample sets obtained at the same observational sites. However, almost none of the chemical component in the WSM (besides WSOC) is shown in the manuscript, only referring to the prior papers. The authors should show the amount of each inorganic component (at least SNA) relative to that of WSOC and should add more discussion on that data. Otherwise the discussion is very weak regarding the interaction between WSOC and inorganics, and their contribution to the hygroscopicity.**

Detailed chemical characterizations are not available for every study, but we have added the available data. Fortunately, for the SPL study, the focus of our analysis, there is data. As the referee notes, this information is critical in our effort to isolate the effect of WSOC on hygroscopicity

*Changes in manuscript: We have added tables describing the chemical composition, where it is available. These tables make table 2 redundant and it has been removed.*

Available WSM composition data has been included in Table 2, including SNA.

**(5) Table 1: It is not clear how the authors estimated "sulfate cleaning efficiency."**

Cleaning efficiency is one minus the ratio of the concentrations of sulfate in the WSOC and WSM samples, given as a percentage. Sulfate concentration was measured using IC.

*Changes in manuscript: We clarify this in our expanded description of the chemical analysis and extract preparation.*

Page 6, lines 13-15, describe the basis for estimating sulfate cleaning efficiency.

**(6) The authors have used the term "highly soluble WSOC" in several parts of the manuscript (e.g., P.10 L.20, P.11, L.7). What is the definition of this "highly soluble WSOC," and what is the difference between this term and "isolated WSOC" which the authors measured?**

This point is well taken. We intend no particular category of "highly soluble WSOC," distinct from isolated WSOC. It is an imprecise way of reminding the reader that WSOC is more soluble than many inorganic WSM species.

*Changes in manuscript: We have removed or reworded each instance of this usage.*

Each use of 'highly soluble' has either been replaced or supplemented with "low-DRH."

**(7) The authors should add more details on the site descriptions: characteristics of each site (rural, mountain, forest, relative influence of anthropogenic vs. biogenic sources, etc.) should be listed maybe in Table 1 or in an additional table. Also why were those sites selected or appropriate to study the hygroscopicity of WSOC? It is really difficult to understand the major difference among the sites for general readers.**

*Changes in manuscript: Basic site details have been added and more discussion of the sites added to Section 2.1.*

P.5, lines 6-15: The sites have been described in greater detail, including location, general character, and basic influences.

**(8) In figures 4-6, the authors Most of readers may not be interested in the sample number, but interested in the exact difference among the samples in terms of their chemical characteristics.**

The numbers were originally intended to allow comparison with prior publications but are now useful for referencing tabulated chemical characteristics.

Table 2 has been added and relies on the sample numbers. The numbers are also useful for correlation with earlier studies and with the data supplement.

**(9) Section 3.6: Discussion on possible effects of aerosol shape is too speculative. Again, the authors only refer to the previous works and do not show any supporting data to convince readers of their hypothesis. For example, if the authors assume a shape factor other than 1, then how can this support the discussion here?**

This criticism is well received. We have made changes as discussed in our response to comment 1.7 above.

In response to this criticism, we have largely rewritten Section 3.6 to emphasize the conclusions of other researchers who have detected similar behavior, and move away from speculation.

(10) In Table 1, "SO4" should be "SO $4$ $2-$" or "sulfate"

This change has been made to Table 1.

**Referee Comment III:**

**The study reports hygroscopic growth factors, kappa and CCN activity of water-soluble aerosol components in the form of re-aerosolized liquid samples from filters originally collected in National Park locations in the US. The methods, in particular the separation of WSOC from WSM, and the separate analysis of WSOC hygroscopicity, are novel and very interesting to the community. The results show that WSOC has large effects on the hygroscopic behaviors of mixed organic/inorganic aerosols. The paper is well-written and the topic is very relevant.**

We are grateful for Referee III's thoughtful review of this manuscript. We believe that we have answered each comment and improved the discussion paper significantly.

**The following points should be addressed before publication:**

**General comments 1) The complex experimental setup and measurement program deserve a more detailed description, in particular concerning uncertainties and the exact measurement program. What were the ranges of uncertainty and stability of the various RH settings and flow rates? How were the TDMA data inverted and growth factors determined? What were the uncertainties in the determined activation diameters? The section on TDMA and SMPS-CCNC operation would also profit from a better links between the text and Figure 1: What were the setpoints of size and RH at which points in the setup and in time for which measurement series?**

We have expanded the methodology section in response to this and similar comments above, with special attention to these questions.

> The description of the experimental setup given in Section 2 has been greatly expanded in response to this and similar comments. Measurement uncertainties for hygroscopicity and CCN analysis are given on page 9, lines 15-28; for the chemical analysis on page 6, lines 18-19. The program of chemical analysis is also described on page 6, lines 9-25. The TDMA and CCN-SMPS measurements are elaborated in Section 2.5.
>
> TDMA operation is described in more detail on page 7, lines 19-32, including links with Fig. 1 and the relevant size and RH setpoints.
>
> The processing of TDMA data is described on page 8, lines 19-26.

**2) More detail should also be provided on the handling and chemical analysis of the filter samples. How many filters, and which days were combined for each re-aerosolized sample? How might aerosol properties have changed over the course of the days combined into each sample? Also, it appears that some of the filter samples were taken as long as ten years ago - when were they extracted, and when were the laboratory measurements done? Can it be ensured that there are no artefacts from storage due to evaporation or other processes? In the results, fractions of WSOC and OC are re- ported, as well as WSOC retention percentages. How was this assessed, was there an independent WSOC measurement? How (and when) was OC measured? It would also be**

**much more reader-friendly to present the inorganic composition of the combined samples in this paper, rather than (more or less explicitly) referring to earlier studies (do they feature the same combination of filters to larger samples?).**

We have attempted to respond to each of these questions, but do not reach the variation in aerosol properties over the course of days. Adequate treatment of aerosol sources is beyond the scope of this (quickly lengthening) paper.

> *Changes in Manuscript: We have added more details on the capture and preparation of the WSOC extracts. The basic sample information (how many days were combined, etc.) are already included in the supplement. We have added some limited discussion of the preservation of the samples and the timeline of the laboratory studies. We also have expanded the description of the chemical analysis (OC and WSOC). Finally, as we note in response to Comment 4 of referee 2, we have included tables with the available composition data. This includes the detailed inorganic composition of the Storm Peak Lab samples, the focus of our analysis.*

> More detail on filter handling and chemical analysis is given now given in Section 2.2. Details of which daily filters are combined in each sample are included in the Data Supplement, which is now referenced on page 6, line 1.

> The timeline of analysis is given more attention (e.g. page 6, line 27, which describes the time between extraction and hygroscopicity analysis). The year-to-year stability of extracts is defended on page 6, line 28.

> The chemical analysis procedures and assessment WSOC concentration are described on page 6, line 15-23.

**3) The discussion on the enhancement of hygroscopic growth through WSOC could, and should, be more quantitative. "Enhancements" would be more obvious if measured values were compared to reference values - either of measured hygroscopic properties of the inorganic components (not just the total WSM) or of theoretical values/theoretical growth curves of inorganic salts in the Figures. See more specific comments below.**

This comment is shared by each referee and the point is well taken. Our approach is detailed above in the response to the first comment of referee 1. In sum, we have followed this suggestion fairly closely by incorporating theoretical growth curves or inorganic compounds drawn from the E-AIM model.

> This approach has been largely adopted, resulting in a reworking of Sections 3.3-3.5 in the manuscript. A detailed response is given above in response to the first comment, but it is mostly duplicative of the actual changes and discussion in the manuscript.

**4) The discussion of impact of particle shape needs substantiation. It seems a stretch to invoke two different mechanisms of particle shape changes as an explanation for both the deliquescence and efflorescence branch of the growth curves. The collapse of particle structures upon hydration is a known effect for aerosol types such as fresh com-**

**bustion agglomerates, but it is a far less obvious thought for a re-aerosolized WSOC sample. How do the authors know that the atomized aerosol is "irregularly shaped"? A discussion of the growth factor uncertainties deriving from the experimental setup and from GF determination from the DMA2 size distribution should be given. Also, why are the GF<1 not showing in the GRSM GF curve in Figure 2?**

This critique is well taken and is shared by each reviewer; a full response is given for comment 1.7. Briefly, we have reworked the shape-factor discussion to focus on prior examples in the literature and practical effects on our results. While we have no definitive basis in our measurements for this conclusion, there are several groups that have reported and investigated similar behavior. Most notably Mikhailov et al., 2009, claim that spray drying of oxalic acid can produce aerosol with void fractions approaching 40%.

In response to this and similar comments, Section 3.6 has been reworked to focus on earlier examples reported in the literature.

The uncertainties associated with growth factor are now discussed on page 9, lines 21-28 and page 11, lines 15-18.

The growth curves in Fig. 2 are normalized to reflect the minimum size recorded, as discussed on page 17, lines 11-13 and page 10, lines 14-16.

Gysel, M., E. Weingartner, et al. (2004). "Hygroscopic properties of water-soluble matter and humic-like organics in atmospheric fine aerosol." Atmos. Chem. Phys. 4(1): 35-50.

Boreddy, S. K. R. and K. Kawamura (2016). "Hygroscopic growth of water-soluble matter extracted from remote marine aerosols over the western North Pacific: Influence of pollutants transported from East Asia." Science of The Total Environment 557–558: 285-295.

Mikhailov, E., S. Vlasenko, et al. (2009). "Amorphous and crystalline aerosol particles interacting with water vapor: conceptual framework and experimental evidence for restructuring, phase transitions and kinetic limitations." Atmos. Chem. Phys. 9(24): 9491-9522.

Detailed comments:

**p.4, Section 1.1: Please give more details on those locations and sampling sites.**

As we note in response to comment 2.7, we have added details to Section 2.1, page 5, lines 4-15.

**p.6, line 20: Why was this exact dry diameter chosen?**

The diameter was chosen to correspond to the peak of the size distribution generated by the atomizer; these were very dilute solutions.

Page 7, lines 21-22 clarify this choice.

**p.6, line 24: How often is "periodically"?**

*Changes to the manuscript:  We specify that extensive calibrations were done at the outset of the laboratory portion of these measurements, while RH calibrations were conducted at the beginning and at least once more over the course of the measurements for each study (a period of ~10 days).*

Page 9, lines 12-14, clarify calibration frequency.

**p.7, lines 14-16: This should be described earlier, along with more details on the measurement methods and results of the inorganic compounds.**

We have added this description to the methodologies section (p.6, lines 9-25) and added the inorganic composition in table 2.

**p.7, line 23: Please specify here: which one is the winter study? It is not reader-friendly to have him/her leaf back to the study description, find the abbreviation that refers to the winter study, and then re-locate that abbreviation in Figure 2.**

*We have incorporated this suggestion.*

GRSM Winter has been added in a parenthetical to make this clear.  P.10, line 6.

**Figure 2: a) dark blue and dark green, as well as light blue and light green are hard to distinguish. How about a different symbol but the same color for WSM and WSOC of the same location?**

*We have made these modifications to Figure 2.*

We have modified the symbols in Fig. 2 and attempted to improve the clarity.  We have kept the color scheme for the sake of matching other figures.

**It would also be nice to call "GRSM I"/"GRSM II" "GRSM summer" and "GRSM winter", to spare the reader repeated leafing back for which one is which.**

We have made this change to Figure 2 and throughout the manuscript.

**b) This presentation of activation diameters should be commented on in more detail in the text. Alternatively, it could be dropped.**

We have added some description to the methodology section (p.8, line 28) and dropped this from the caption.

**p.8, line 19: what kind of change classifies as "minor", what as "large"? Please expand (or add a reference) on how a "large" change in kappa indicates a phase change.**

We have rephrased this description to use the more familiar and descriptive word 'abrupt.'

p.8, line 30-31: In Figure 3, GRSM I starts at around 0.3 -I would not call this "near zero".

P.11, line 21: This was a mistake and has been corrected to reference SPL and MORA samples.

**p.9, line 8: "complementary enhancement" as compared to what? The inorganic components alone? (Below or above efflorescence?) For this, a reference hygroscopicity value (measured or calculated) of the isolated inorganic components should be given. What is currently shown in Figure 3 is just that total WSM has a higher kappa than the WSOM, which is not surprising. The paper would improve substantially if this analysis could be more quantitative.**

Sections 3.3-3.5, including this discussion, have been largely reworked in an attempt to quantify this analysis.

**p.9, line 6-7: These chemical compositions should be shown in a figure.**

We have incorporated this composition data into Table 2.

**p.9, line 20: Where and how is this shown?**

This claim is more strongly supported now, and more clearly and correctly stated in the reworked analysis of this section.

**p.9, line 26: Where is this argument going?**

We clarified this discussion to reflect the more quantitative approach adopted in response to these comments.

[revised manuscript text omitted]